



# Spatial and temporal variability of snow accumulation for the South-Western Greenland Ice Sheet

Achim Heilig[1,2], Olaf Eisen[3,4], Martin Schneebeli[2], Michael MacFerrin[5], C. Max Stevens[6], Baptiste Vandecrux[7], and Konrad Steffen[8]

[1]Department of Earth and Environmental Sciences, LMU, Munich, Germany
[2]WSL Institute for Snow and Avalanche Research SLF, Davos Dorf, Switzerland
[3]Alfred Wegener Institute Helmholtz-Centre for Polar and Marine Research, Bremerhaven, Germany
[4]Department of Geosciences, University of Bremen, Bremen, Germany
[5]Cooperative Institute for Research in Environmental Sciences, University of Colorado, Boulder, CO USA
[6]Department of Earth and Space Sciences, University of Washington, WA USA
[7]Department of Glaciology and Climate, Geological Survey of Denmark and Greenland, Copenhagen, Denmark
[8]Swiss Federal Research Institute WSL, Birmensdorf, Switzerland

**Correspondence:** Achim Heilig (heilig@r-hm.de)

**Abstract.** The Greenland ice sheet (GrIS) has experienced significant changes in recent decades. Data confirming those changes are derived from remote sensing, regional climate models (RCMs), firn cores and automatic weather stations (AWSs) on the ice sheet. Data sources comprise different extents in area coverage. While remote sensing and RCMs cover at least regional scales with an extent ranging from 1–10 km, AWS data and firn cores are point observations. To link such regional scales with

point measurements, we investigate the spatial variability of snow accumulation within areas of approximately 1–4 km$^2$ and its temporal changes. At three different sites of the southwestern GrIS (Swiss Camp, KAN-U, Dye-2), we performed extensive ground-penetrating radar (GPR) transects and numerous snow pits. In dry snow conditions, radar-measured two-way travel time can be converted to snow depth and snow accumulation if the density is known. Density variations per site for snow pits within distances of up to 1 km are found to be consistently within $\pm 5\%$. GPR transects were further filtered to remove small

scale surface-related noise. The combined uncertainty of density variations and spatial filtering of radar transects is at 7–8% per regional scale. To link point observations with regional scales, we analyze for spatial representativeness of snow pits. It occurs that with a probability of $p = 0.8$ (KAN-U) to $p > 0.95$ (Swiss Camp and Dye-2), randomly selected snow pits are representative in snow accumulation for entire regions with an offset of $\pm 10\%$ from arithmetic means. However, to achieve such high representativeness of snow pits, it is required to average snow depth for an area of at least 20 m x 20 m. Interannual

accumulation pattern at Dye-2 are very persistent for two subsequent accumulation seasons with similarity probabilities of $p > 0.95$, if again an error of $\pm 10\%$ is included. Using target reflectors placed at respective end-of-summer-melt horizons, we additionally analyzed for occurrences of lateral redistribution within one melt season. In this study, we show that at Dye-2 lateral flow of meltwater cannot be evidenced in the current climate. Such studies of spatial representativeness and temporal changes in accumulation are inevitable to assess reliability of the linkage between point measurements and regional scale data

and predictions, which are used for validation and calibration of remote sensing data and RCM outputs.



# 1   Introduction

Numerous recent studies have documented a continuous mass loss from the Greenland ice sheet (GrIS) (e.g., Shepherd et al., 2012; Velicogna et al., 2014; Khan et al., 2015; van den Broeke et al., 2016; Sørensen et al., 2018; Mouginot et al., 2019)

using remote sensing data and/or estimates from model simulations. From 1980 to 2018, mass loss from the GrIS increased by a factor of six (Mouginot et al., 2019), and over the last two decades the major mass loss process has changed from solid ice discharge to surface mass balance (SMB) related processes (van den Broeke et al., 2016). SMB can be regarded as sum of snow accumulation and lateral redistribution by sublimation, wind and runoff (with positive and negative sign). Over most of the GrIS, net accumulation is the dominating factor for SMB (Koenig et al., 2016), while negative trends in SMBs are related

to surface melt and runoff (Vaughan et al., 2013). Despite their importance for the GrIS mass balance, SMB estimates remain a major source of uncertainty in ice-sheet mass-balance calculations (van den Broeke et al., 2009). This is because surface mass fluxes, such as snowfall and melt, cannot be measured by remote-sensing technology, and in situ measurements are scarce. Estimates of SMB are usually obtained using in situ measurements in concert with dedicated regional climate models (RCMs), which can introduce significant uncertainties (Vernon et al., 2013): Different scales between in situ observations and

simulations certainly contribute to these uncertainties. The spatial resolution of RCMs and remote sensing data are limited to regional scales (on the order of one to tens of square kilometers), while in situ observations cover point scales (on the order of a few square meters or less). Effects of wind redistribution, for instance, are leveled out for regional scales but can have significant influences at point scales. As a consequence, evaluation and validation of regional-scale data products using in situ data is difficult without knowledge of the spatial extent and representativeness of the point measurements. To date, only a few

studies have analyzed how representative point observations (e.g., snow pits, firn cores, mass-balance-stake readings, automatic weather station [AWS] measurements) are of the surrounding several square kilometers.

Within the last decade several studies have used radar systems (Ground Penetrating Radar [GPR] and Frequency-Modulated Continuous-Wave radar [FMCW]) to quantify accumulation variability in Greenland by tracking internal reflection horizons (IRHs) (e.g., Dunse et al., 2008; Miège et al., 2013; Hawley et al., 2014; Koenig et al., 2016; Lewis et al., 2017). While

those studies aimed to track IRH variability using data from long ground transects of roughly 100 km (Miège et al., 2013) to more than 1000 km (Hawley et al., 2014) length or using airborne radar data (Koenig et al., 2016; Lewis et al., 2017), only one worked to link the point observations from snow pits and cores to the surrounding area (Dunse et al., 2008). For instance, Koenig et al. (2016) used airborne radar data from NASA's Operation Ice Bridge to calculate accumulation rates with a stated uncertainty of 14%, and they compared their results to outputs from an RCM. Radar-derived accumulation results

are compared to two sites with core data, but the locations of those sites data are up to 8 km away from the radar track. It is not possible to identify whether mismatch between the core- and radar-derived accumulations is due to spatial variability or to assumptions in radar-data processing. Other recent studies at the GrIS attempt to relate point observations of melt events within the percolation zone of the GrIS with annual atmospheric patterns (Graeter et al., 2018) or determine the mass of



percolating liquid water and compare percolation depths observed by upward-looking radar (upGPR) with temperature records
in snow and firn (Heilig et al., 2018). In addition, several studies have quantified temporal accumulation variability using ice
core records (e.g., Mosley-Thompson et al., 2001; Vandecrux et al., 2019). Still, quantification of spatial representativeness of
single point measurements for the surrounding square kilometers has only been conducted for one point in western Greenland
so far (Dunse et al., 2008), which implies there is a need to explore uncertainties at local and regional scales. The best means of
resolving these uncertainties are to increase the spatial coverage of direct measurements (Farinotti et al., 2014) and to improve
our understanding of how well point measurements represent a larger area.

With rising global temperatures being amplified in northern latitudes (e.g., Meehl et al., 2012), surface melt increasingly
affects SMB (e.g., Sasgen et al., 2012). Annual accumulation estimations (snapshots in time) from radar data, ice cores and
snow pits are made by determining the mass of a specific layer. Consequently, the sparse validation of RCMs and remote
sensing data is only performed for the time period when in situ data are collected. Melt occurrences have significant effects on
snow and firn layers. Several observations indicate that meltwater percolation can move mass from snow to the underlying firn
(e.g., Charalampidis et al., 2016; Humphrey et al., 2012; Heilig et al., 2018) or even laterally along the surface slope (Humphrey
et al., 2012). It is very unlikely that water percolation and mass redistribution are homogeneous over regional scales. However,
so far, no assessment of the impact of melt on temporal changes in accumulation distribution has been performed for the
percolation zone of the GrIS.

To improve understanding of representativeness of point measurements, the purpose of this work is to relate point scales to
regional scales of one to several square kilometers in area for two sites within the percolation zone of the GrIS for two years
and for one site at the equilibrium line of altitude. Major questions this study aims to answer are: (i) How large is the error in
IRH tracking of radar data? Uncertainties arise from applying density assumptions, the existing surface roughness and applied
sample resolutions. (ii) How representative is a single point measurement of accumulation of a larger area in the context of
spatial variability within that area? (iii) Are accumulation patterns temporally and regionally consistent over several seasons?
(iv) Can a temporally continuous accumulation observation be related to area-wide changes in accumulation or is melt and
water percolation affecting accumulation differently? In other words, does lateral flow affect the accumulation pattern during
melt?

To answer the questions above, we examine snow-pit and GPR data from two sites within the percolation zone of the GrIS
and one site at the equilibrium line of altitude gathered over several field seasons. For each site, we analyze density variability
between measurements from up to six snow pits within an area of 4 km$^2$ made in a single season, process radar transects
of up to 25 km recorded in close proximity to those snow pits, and spatially extrapolate the radar-derived accumulation to
estimate area-wide accumulation variability. For temporal comparisons, we use continuous observations of accumulation and
melt recorded by upGPR (Heilig et al., 2018). Our results show that spatially averaged snow-depth measurements are necessary
to assess accumulation on regional scales.



**Table 1.** Metadata for the five GPR transects analyzed in this study. DoA is date of acquisition.

| Location | DoA | Trace distance [m] | Total length [km] | Antenna frequency [MHz] |
|---|---|---|---|---|
| KAN-U | May 2013 | 1.5 | 15.3 | 800 |
| Swiss Camp | May 2015 | 0.05 | 0.35 | 1600 |
| Dye-2 | May 2016 | 0.5 | 20.6 | 1600 and 600 |
| Dye-2 | May 2017 | 0.5 | 24.9 | 200 and 600 |
| KAN-U | April 2017 | 0.5 | 10.9 | 200 and 600 |

## 2 Methodology

### 2.1 Test site, instrumentation and data processing

We collected radar data along transects at three different locations on the southwestern GrIS over several years (Figure 1, Table 1). The sites were visited in spring of each year (see Table 1). At Swiss Camp (69.5552°N/ 49.36525°W at 1170 m above sea level [asl]) a small transect was measured in May 2015 by towing a GPR trolley on foot. The measurements were triggered by an odometer wheel. Geolocation was only performed for starting and end points of some radar lines, and locations in between are interpolated. The radar data from Swiss camp have 0.05 m trace distance along track. The transects at Dye-2 (66.47785°N/ 46,28564°W, 2120 m asl) and KAN-U (67.0011°N/ 47.02757°W, 1860 m asl) were recorded in time mode and dragging the antennas behind a snow machine. Because small variations in snow-machine speed cause recorded radar traces to be spaced unevenly, the traces are averaged to generate equidistant spacing. The resulting horizontal trace distance is 0.5 m for both Dye-2 transects and the 2017 KAN-U transect. The trace spacing along the 2013 KAN-U transect is 1.5 m because the snow-machine speed was faster. For the Dye-2 and KAN-U surveys, antennas were connected to a GPS receiver for geolocation of the GPR transect.

We used two different units for the recorded five radar transects. At Dye-2 and KAN-U in May 2017, we employed an IDS (Ingegneria dei Sistemi, Pisa, Italy) FastWave control unit with dual frequency antennas. The respective frequencies are listed in Table 1. Radar measurements at Swiss Camp in May 2015 and at KAN-U in May 2013 were conducted using a RAMAC system (MALA Geoscience, Sweden).

All recorded radar traces were processed in a very similar way. In case first arrivals were delayed by more than approximately 2 ns, we started with a correction for the DC shift, applied dewow filtering, followed by bandpass filters adjusted to the respective center frequency of the antennas. We further applied background removals to minimize direct wave influences. For all radar transects, we corrected for divergence losses by gain functions and interpolated to equidistant traces. The zero-crossings of the snow surface reflections were corrected to be at time zero.

The measured quantity of radar transects is the two-way travel time (TWT with mathematical symbol $\tau$) from the transmitter to the reflector and back to the antennas (e.g., Heilig et al., 2018). In dry snow and firn (with two contributing volume fractions $\theta_a + \theta_i = 1$), the wave propagation depends solely on the relation of air ($\theta_a$) to ice volume fraction ($\theta_i$) (e.g., Kovacs et al.,




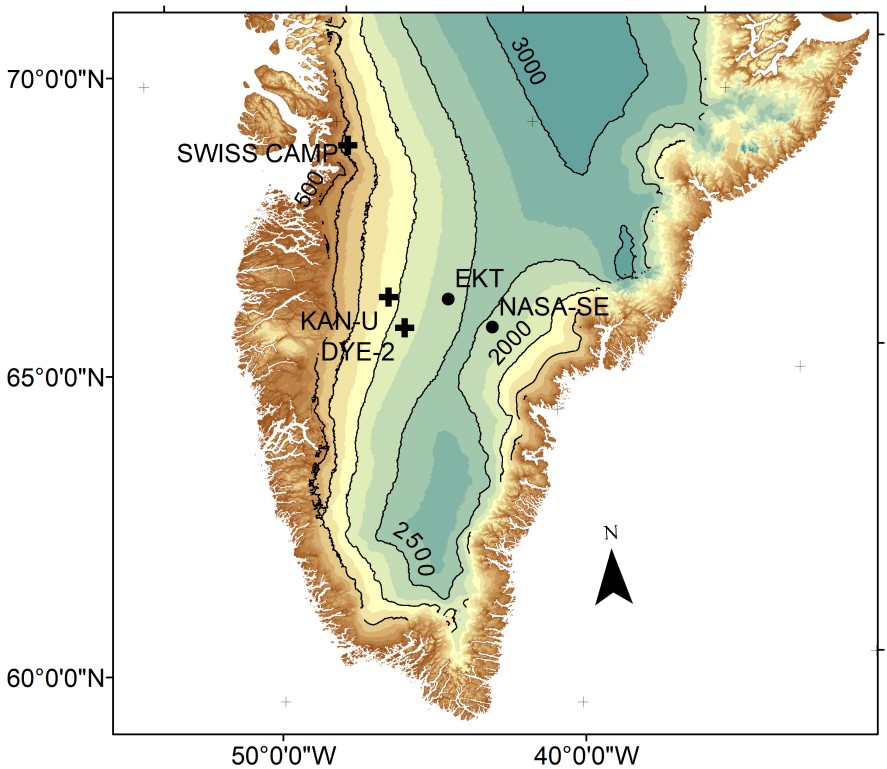

**Figure 1.** Map displaying locations of radar transects analyzed for this study in Southern Greenland (black crosses). The black dots indicate additional locations where snow pits were dug for snow density analysis. The colors are 250 m contour lines and the black contour lines are 500 m intervals. The underlying digital elevation model was generated by Howat et al. (2014).

1995; Mätzler, 1996). Hence, with the snow density ($\rho_s$) measured in snow pits, we can convert from TWT to snow depth ($\overline{L_s}$) and the amounts of bulk accumulation in snow water equivalent (SWE with unit kg/m$^2$ and mathematical symbol $b_s$) using the equation

$$b_s = \overline{L_s}\rho_s \tag{1}$$

with

$$\overline{L_s} = \frac{\tau}{2}\frac{c}{\frac{\rho_s}{\rho_i}(\varepsilon_i^\beta - 1) + 1}. \tag{2}$$

The ice density ($\rho_i = 917$ kg/m$^3$), the speed of light in vacuum ($c$) and the relative dielectric permittivity of ice ($\varepsilon_i = 3.18$) are constants taken from previous literature (e.g., Heilig et al., 2018). The reflections of the previous end-of-melt-season (EMS) horizons are clearly detectable in all radargrams. We relate internal reflecting horizons (IRHs) to depths at pit locations using 120 the measured bulk $\rho_s$. Accordingly, we choose the zero-crossing of the IRHs as the first break of the respective layer. To



**Table 2.** Locations of snow density analyses with with date of acquisition (DoA), number of snow pits (N), mean density ($\bar{\rho}_s$) and density range.

| Location | DoA | N | $\bar{\rho}_s$ [kg/m$^3$] | range [%] |
|----------|-----|---|---------------------------|-----------|
| Dye-2 | May 2015 | 6 | 355.5 | -4 – +2 |
| EKT | May 2015 | 5 | 341.3 | -5 – +4 |
| NASA SE | May 2015 | 2 | 364.5 | -2 – +2 |
| Swiss Camp | May 2015 | 4 | 358.4 | -5 – +5 |
| Dye-2 | May 2016 | 6 | 320.1 | -6 – +4 |
| EKT | May 2016 | 3 | 339.2 | -2 – +2 |
| KAN-U | April 2016 | 4 | 346.0 | -6 – +5 |
| NASA SE | May 2016 | 2 | 369.7 | -1 – +1 |
| Swiss Camp | May 2018 | 3 | 351.3 | -2 – +3 |

identify the EMS horizon of 2015 at Dye-2 in May 2017, we make use of target reflectors that were buried in May 2016 on the 2015 summer horizon. Hence, in May 2017, it was possible to revisit those locations with the radar and unambiguously distinguish between signal reflections arising from the 2015 and 2016 EMS horizons.

However, before applying a constant $\rho_s$ over the entire length of the radar transects, one has to investigate the spatial
heterogeneity in $\rho_s$ over an area of comparable size. To accomplish this, we dug numerous snow pits at Dye-2 in May 2015 and 2016, at Swiss Camp in May 2015 and 2018 and at KAN-U in April 2016. In each pit, we measured the bulk density of the snow from the surface down to the previous season's melt surface. The snow pits were dug at various distances from each other up to 1 km. In addition to locations where we collected radar data, we also investigated spatial variability in $\rho_s$ at two more sites, EKT and NASA-SE (Figure 1). Table 2 displays the numbers of snow pits, the mean density of all pits for that site and year, and the ranges (minimum divided by mean and maximum divided by mean) in percent. To process the radar data
collected at Dye-2 in May 2017, we use density data from firn cores to calculate radar wave speed between the summer 2016 and summer 2015 horizons. Snow temperature measurements ensured dry and subfreezing conditions.

For all three sites, long term meteorological observations exist. To discuss the meteorological conditions at each site, we use wind data from the GC-Net stations (Steffen and Box, 2001) at Dye-2 (September 2011 to May 2018, with gaps in between) and
at Swiss Camp (May 2016 to May 2017) and the PROMICE station (van As et al., 2011) at KAN-U (April 2009 to September 2016).

## 2.2 Transect data analysis

TWT data are influenced by vertical sampling and small-scale surface roughnesses. Wind-induced surface features, such as sastrugi, appear in 2-D radar transects as discontinuous, erratic noise. Ideally, we would have performed radar surveys on high-
resolution grids (i.e. with spacing smaller than the characteristic length of the features) to spatially extrapolate such features to





the non-surveyed areas. However, it was not possible to conduct such high-resolution surveys in the one to two days available at our sites. Instead, we apply spatial smoothing to minimize artifacts from vertical sampling and to remove wind-induced surface-feature noise.

The time sampling of the recorded GPR transects range from 0.05 ns per sample (Swiss Camp 2015) to 0.24 ns per sample (Dye-2 2017), corresponding to approximately 0.006 m and 0.028 m per sample respectively. For the longer transects at KAN-U and Dye-2 (Table 1), the vertical sampling is constantly coarser than 0.1 ns/sample. As displayed in Figure 2, the raw radar data for these transects are continuously fluctuating by $\pm 1$ sample (corresponding to roughly $\pm 3$ cm). Such effects are caused by amplitude clipping of the signal response and uncertainties of the zero-crossing as consequence of the vertical sampling. For each radar trace, we picked consistently the first strong positive half cycle and shifted the first break upwards to match

the zero-crossing. However, due to a vertical sample intervals of 0.25 ns, it is likely that the strongest amplitudes shift by 1–2 samples for consecutive radar traces. To reduce effects caused by the amplitude shifts, in our (lower resolution) KAN-U and Dye-2 data, we applied a Savitzky-Golay filter (Savitzky and Golay, 1964) with frame length of 20 m and polynomial order of 3 (Figure 2, red line). At Swiss Camp with the much finer vertical sample interval, it is adequate to filter with 1 m frame length to reduce clipping and zero-crossing uncertainties.

At Swiss Camp, where we surveyed on a sub-meter grid, we are able to analyze small scale accumulation variability directly. For the other two sites, however, the transects were several kilometers in length and not in a regular grid. To enable quantitative geostatistical extrapolation over areas not surveyed with the radar, it is necessary to remove small-scale surface roughness from the data. With a horizontal sampling resolution of 0.5 to 1.5 m, the variability in the radar-derived snow depth is dominated by surface-wind features such as dunes and sastrugi. As exemplarily demonstrated in Figure 2 (red line), variability has an

average wavelength of roughly 20–30 m and an amplitude of roughly 10 cm on average. To minimize surface roughness, we again employ Savitzky-Golay filtering. We search numerically for filter frame lengths for which the average standard deviation within a 20 m radius around each radar trace is 1 cm or less; a smoothing length of 20 m has been used by other recent studies dealing with large scale GPR transects 20 m distances as well (e.g., Lewis et al., 2019). The resulting filter frame lengths range from 135 m (Dye-2, May 2016) to 210 m (KAN-U, May 2013), which allowed the removal of high frequency variations with

an amplitude of about $\pm 0.1$ m (Figure 2 green line). We use the smoothed data for spatial extrapolation.

## 2.3 Spatial extrapolation

In order to analyze accumulation patterns over a larger area, it is necessary to extrapolate the data gathered along the radar transects. One radar trace provides a single depth estimate to a specific reflector. Combining GPR-derived snow accumulation transects with geostatistical techniques, however, is a powerful method to model spatial occurrences of continuous subsurface

features. Similar combinations of geophysical and stochastical techniques have been used in previous research (e.g., Rea and Knight, 1998; Tercier et al., 2000). The benefit of radar data is that numerous data pairs for a wide range of measurement distances are recorded enabling more constraint experimental variograms. Webster and Oliver (2007) state that sample size is directly related to the precision of variogram estimates. Before extrapolation of a data parameter, the data must fulfill several prerequisites. First the recorded data have to be continuous and for a specific distance spatially correlated (e.g., Rea and



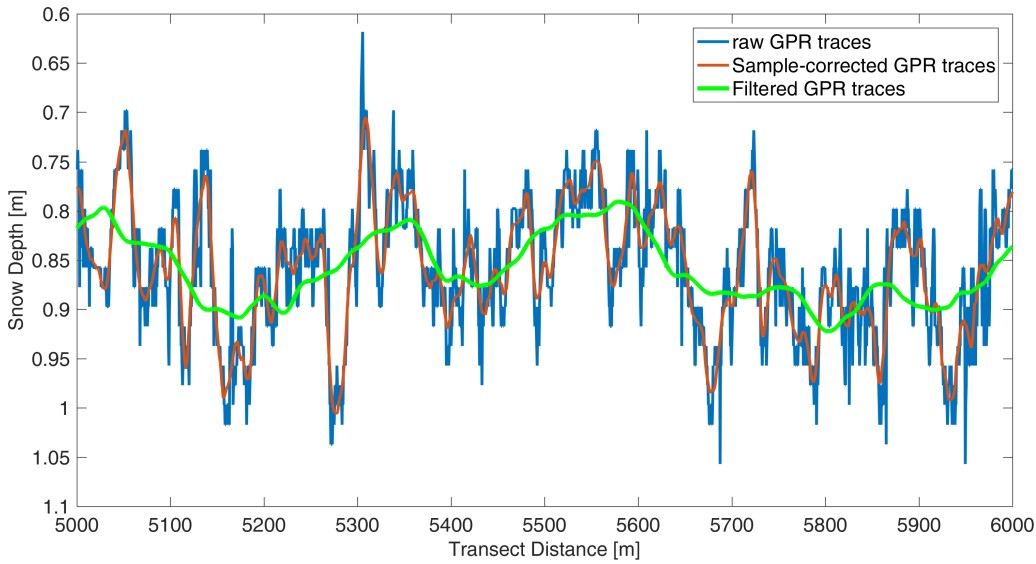

**Figure 2.** A 1 km section for radar-derived snow depths for the end-of-melt season 2016 horizon of the radar transect from May 2017 at Dye-2. Displayed are the raw first breaks (blue line), after being corrected for sample uncertainties (red line) and the final product being used to assess spatial variability for an area of several square kilometers (green line).

Knight, 1998). Snow-accumulation at the analyzed sites are continuous (there are no gaps in accumulation in between) and have intrinsic stationarity, i.e. that the expected mean and variance of the data are invariant in space. To ensure the stationarity, we analyzed X- and Y- directions separately checked for trends with direction and subtracted them before further analysis. At DYE-2 and KAN-U, we discovered accumulations trends in both X- and Y- directions while at Swiss Camp a one dimensional trend was found.

Figure 3 displays the probability distributions of all five radar transects. If the distribution (plotted crosses) follows the straight line, the data are normally distributed. At least a realm of 10–80% of data match normality for all five GPR transects. Kurtosis and skewness for all transects are smaller than ±0.3. We thus consider our data adequately normal to not require data transformation prior to geostatistical analysis.

For spatial extrapolation, we used the Geostatistical Analyst toolbox in ArcGIS10.4.1. Because our data are normally distributed with just one variable (snow accumulation), we use ordinary kriging, which is the most robust and most commonly used method (Webster and Oliver, 2007). Despite the trend removal, anisotropy of the covariance in horizontal directions is still present in all of the longer transects. Hence, we modeled variograms with various distances for different directions. Again, the much smaller transect at Swiss Camp is an exception and can be modeled simply by an isotropic variogram. The geostatistic parameters used for ordinary kriging of each transect are presented in Table 3 together with accuracy assessments through prediction errors. At Dye-2 a spherical variogram model provided highest prediction accuracies while at KAN-U and Swiss Camp, the usage of stable variogram modeling resulted in lowest mean prediction errors (values at 0 kg/m$^2$) and best RMS



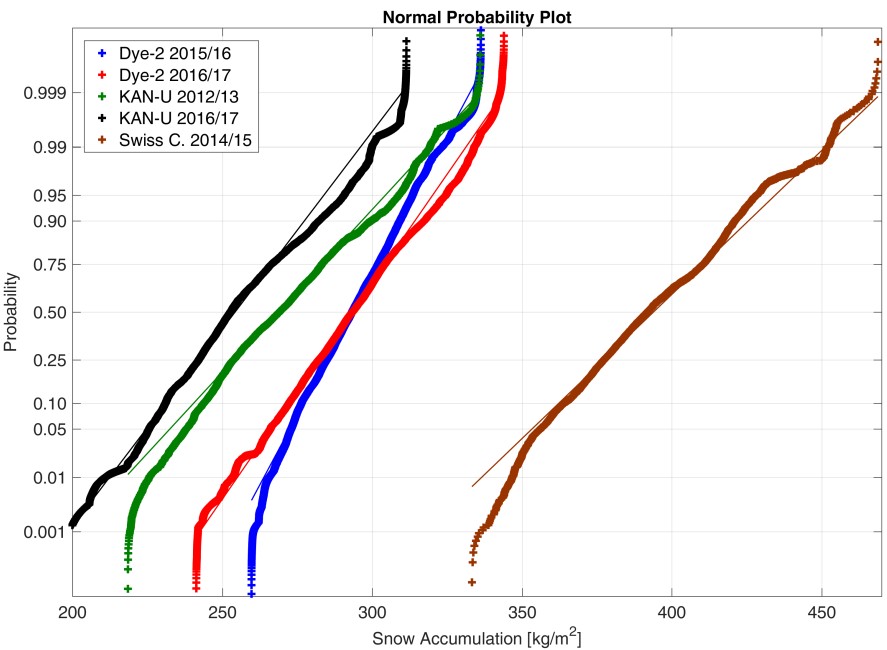

**Figure 3.** Normal probability plots displaying deviations from a standard normal distribution (straight line with corresponding color). The crosses plot the empirical probability versus the data value for each radar-determined SWE value. We display smoothed transects values for KAN-U and Dye-2 and solely vertical sampling corrected values for Swiss Camp.

**Table 3.** Kriging results with description of variogram ranges for the major and minor axis used in the variogram modeling, the resulting mean prediction error (pred. err.) and the root mean square (RMS) standardized prediction error.

| Location | Anisotropy | range major/minor [m] | Mean pred. err. [kg/m$^2$] | RMS standardized pred. err. |
|---|---|---|---|---|
| KAN-U 2012/13 | y | 274/ 91.5 | 0.01 | 1.02 |
| Swiss Camp 2014/15 | n | 3.8 | 0 | 1.11 |
| Dye-2 2015/16 | y | 126/ 80 | 0 | 0.83 |
| KAN-U 2016/17 | y | 96.5/ 63 | 0 | 1.09 |
| Dye-2 2016/17 | y | 156/ 52 | 0 | 1.33 |

standardized prediction offsets (values at 1). The presented variogram ranges in Table 3 represent the direction-wise major extrapolation range. Nugget effects (description of the measurement errors) are small with values far below 5 kg/m$^2$ for all transects. Our kriging outputs have a spatial resolution of 20 m by 20 m for the larger transects and of 0.1 m by 0.1 m for Swiss Camp.

To describe data distribution and assess spatial representativeness, we calculated scaled accumulation values ($b_{s,N}$) and scaled cumulative probability distributions. $b_{s,N}$ is simply the result of the respective kriged accumulation value divided by the





mean kriged accumulation per site and campaign, $b_{s,N} = \frac{b_s}{\bar{b_s}}$. In the following, data distributions are displayed as box plots with the whiskers set to the 5% and 95% percentiles respectively. Whether any randomly located point measurement such as a snow

pit would be representative for the entire extrapolated area is determined using the recorded radar traces. We average all radar traces within a radius of 1 m around each radar trace (which represents a standard pit size) and scale this data point by the mean of the kriged output for the same campaign. Data distribution for each campaign including filtered and sampling-corrected data (see Section 2.2) are presented to describe offset dependencies. At KAN-U for the 2012/13 data, we increased the assumed pit size to an area with 2 m radius because of more sparse horizontal data resolution (1.5 m in between traces). Corner locations

of radar transects with less than four (three for KAN-U 2012/13) neighboring traces within the respective search radius are excluded.

## 3   Results and Discussion

We first discuss errors associated with converting radar-measured TWT to accumulation because understanding these errors is essential for assessing how representative a single point observation, such as a snow pit, is of a larger area; we present that

assessment in Section 3.2. We then analyze accumulation-pattern persistence at Dye-2 and KAN-U, where we collected radar data for two accumulation seasons. Finally, we investigate whether seasonal changes in accumulation due to melt and liquid-water percolation have major effects on the accumulation pattern. Such effects could be caused by strong lateral differences in melt or lateral flow of meltwater. In the following, to distinguish between offsets, deviations from mean and data distribution, we will describe offsets, deviations and uncertainties with values given in percentage (%) and data distribution as probability

values of 0–1.

### 3.1   Error in travel time to accumulation conversion

The conversion from TWT to snow depths using a mean ice volume fraction for entire transects implies errors. It is therefore important to determine the spatial variability in density within the respective area. Table 2 presents snow-pit data from our three study sites and two additional sites. The data were collected over three years, and the distances between pits ranged from

a few meters up to 1 km. The range in density variation from $\bar{\rho}$ in Table 2 - independent of distances in between pits - does not exceed -6 to +5% for nine snow pit campaigns in total, at five different locations for the southwestern GrIS. Calculated range averages for the last column in Table 2 are -3.7 to +3.1%. We thus consider ±5% variation in average density to be a good estimator of uncertainty within areas of several square kilometers for these regions. This corresponds well with observations by Proksch et al. (2016), who derived a mean measurement uncertainty for density of 2–5%.

Uncertainty in $\rho_s$ results in only a small uncertainty in the derived $\overline{L_s}$: $\rho_s$ factors into the conversion of $\tau$ to $\overline{L_s}$ as a fraction within the denominator (Equation 2). For our measured TWTs, a ±5% variation in $\rho_s$ leads to a 0.7–1.4% uncertainty in $\overline{L_s}$ for bulk $\rho_s$ values of 200–450 kg/m³. Additional uncertainty in $\overline{L_s}$ is introduced by the smoothing applied to the larger transects. The average RMS deviation in snow depth of the smoothed transects from the sample-corrected transects at Dye-2 and KAN-U is 4.5 cm (5–6%). Combining the errors due to smoothing of radar traces and using a mean density for processing radar



transects with observed $\rho_s$ variations using Equation 1 leads to an average uncertainty in SWE of 7.0–7.9%. This uncertainty
is significantly smaller than discrepancies between RCM simulations and Operation IceBridge airborne radar determinations
(16%) (Koenig et al., 2016) and smaller than measured relative standard deviations in density observed within the same study
(12%). However, to increase the robustness of accumulation estimates and to decrease effects of spatial extrapolation, we
consider an estimated maximum uncertainty of 10% in SWE determined from radar data as a conservative estimate for regional
catchments of size of 1–5 km$^2$.

### 3.2 Spatial representativeness of point accumulation determinations

#### 3.2.1 Swiss Camp

Figure 4a shows the measured radar grid and area-wide snow accumulation predicted by kriging for Swiss Camp. At Swiss
camp in May 2015, we measured a transect length of roughly 350 m with along-transect resolution of 5 cm and transect lines
separated by 60 cm. Radar data were only filtered to remove sample-related noise (see Section 2.2), which allow us to analyze
small scale variabilities in SWE within 10 cm grid cells. The arithmetic mean of SWE within the analyzed area in Figure
4a is 393 kg/m$^2$ with a standard deviation of 28 kg/m$^2$ (7.1%). Within the northeasterly part of the presented accumulation
distribution (Figure 4a), we find above average accumulation. Along the longer transect lines (from south to north), there
are several spots with below average accumulation. Since the extrapolation was performed in accordance to the observed
variogram range without boundary conditions being set (snow accumulation outside the measured grid existed, we just do not
have information for it), it is impossible to identify minimums and maximums as artifacts or actual variability patterns outside
the grid lines. However, the observed minimums in SWE along the south-north transect lines at regular distances between
8–10 m are likely wind-generated surface features. Prevailing wind direction is from the East with low variations (Figure A1a).
Along the wind direction, the interpolated area range (East - West) does not exceed 21 m, which is less than the wavelength
of the variability pattern observed at Dye-2 (Figure 2, Section 2.2). However, for the cross-wind direction, a wavelength of
8–10 m for dune dips seem to be apparent at Swiss Camp.

Figure 4b displays the scaled SWE data distribution through box plots. The median (red horizontal line), interquartile range
(IQR framed by the blue box), 5% and 95% percentiles (whiskers) and values outside a distribution of $p = 0.9$ (red crosses)
are displayed. Similar to the recorded radar data (Figure 3), a large proportion of extrapolated SWE ($p > 0.9$) follow a normal
distribution. In addition, arithmetic mean, median and mode for this data distribution at Swiss Camp are very similar ($\bar{b}_s =$
392.5 kg/m$^2$, $b_{s,med} = 391.4$ kg/m$^2$, $b_{s,mod} = 381.2$ kg/m$^2$) indicating symmetric data distribution as well (Fahrmeir et al.,
2011). Normal distribution, hence, symmetric data distribution allow direct derivation of distribution probabilities. For instance,
the standard deviation for scaled SWE in Figure 4b is 0.07 which means that $\pm 7\%$ deviation comprise $p = 0.68$ of data.
However, there is a slight difference in whisker lengths (5% percentile at $b_{s,N} = 0.89$, 95% percentile at $b_{s,N} = 1.13$) indicating
a small shift towards higher values and asymmetry for the data distribution tales. Skewness of this data distribution equals
to 0.42. However, $p = 0.86$ of data are within the given $\pm 10\%$ uncertainty for the entire surveyed area. Consequently, the



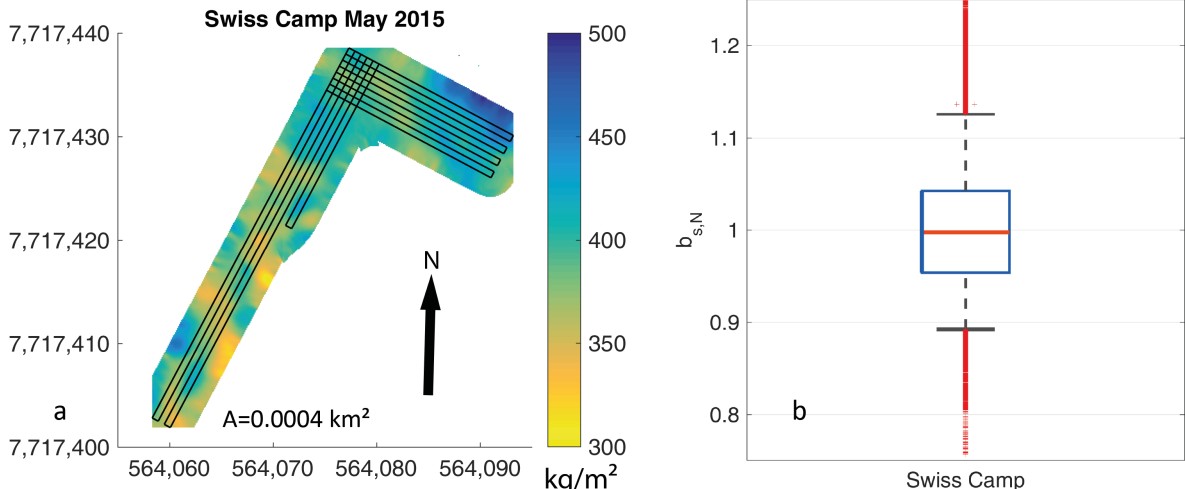

**Figure 4.** (a) Kriging results for the small radar transect at Swiss Camp for snow accumulation. The black lines display the recorded radar traces and the black arrow indicates geographic north. A total area of 400 m$^2$ can be covered by the applied spatial extrapolation. (b) Box plot displaying scaled data distribution ($b_{s,N}$) of kriged output with the red horizontal line showing the median, the box framing the interquartile range and the whiskers displaying the 5% and 95% percentiles. Outliers are shown as red crosses. The coordinates in (a) are given in UTM with datum WGS1984.

presented data distribution in Figure 4b indicates that with a probability of $p = 0.86$, the kriged 10 cm by 10 cm grid points are within 353–432 kg/m$^2$.

We use the recorded radar traces to numerically analyze how representative any pit location of the ∼400 m$^2$ area would be.
As described in Section 2.3, we define a search radius of 1 m around each radar trace, which corresponds in area to a regular pit size in the field. Radar-derived SWE values are averaged within the search radius. Results are plotted as scaled cumulative probability plot (Figure 5). Our analysis shows that $p > 0.95$ of pit locations would provide SWE values within ±10% of the arithmetic mean for those  400 m$^2$ area at Swiss Camp.

### 3.2.2   Dye-2 and KAN-U

For the much longer radar transects at Dye-2 and KAN-U, we filtered out wind-induced surface variabilities of the radar traces to increase spatial extrapolation with enlarged variogram ranges from $10 – 30$ m to $50 – 270$ m (Table 3). Such filtering implies spatial smoothing of surface roughnesses, which could be performed in the field by extensive snow-depth probings. Later in this section, we present comparisons for spatial representativeness of filtered and non-filtered GPR data. In 2016 and 2017, the radar transects were designed to follow the prevailing wind direction to better assess systematic inhomogeneities for Dye-2
and KAN-U in 2017 (see Figures 6, 7 and A1b and c).

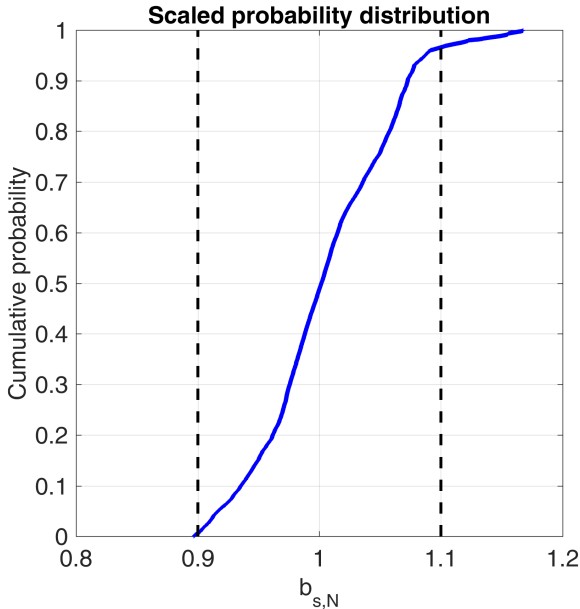

**Figure 5.** Scaled cumulative probability distribution of radar-derived SWE within any subset with 1 m radius of the GPR transect at Swiss Camp. The dashed vertical lines represent the determined uncertainty range of $\pm 10\%$.

At Dye-2, we recorded 21 km of continuous radar data in May 2016 and 25 km in May 2017. This results in geostatistical predictions of snow accumulation over an area of 2.4 km$^2$ for 2015/16 and almost 4 km$^2$ for 2016/17 (Figure 6a and b). The arithmetic mean for SWE in May 2016 is 293 kg/m$^2$ with standard deviation $\sigma = 11$ kg/m$^2$. 2016/17 results in very similar values with a mean SWE of 296 kg/m$^2$ and $\sigma = 15$ kg/m$^2$.

The box plots in Figure 6c represent the same quantiles as in Figure 4b. Data distribution for Dye-2 in 2015/16 is very homogeneous with an IQR of only $\pm 2.5\%$. The whiskers for the same year reach $\pm 6\%$. Hence, SWE in May 2015/16 varies only little with $p > 0.9$ of data within the error margins of $\pm 10\%$. Since already more than 95% of radar-derived SWE follow a normal distribution (Figure 3), extrapolated SWE have a very high distribution symmetry as well. We observe slightly less homogeneity in the subsequent year at Dye-2. Here, the IQR increases to $\pm 3\%$, with 5% and 95% percentiles being slightly

below the error margins of $\pm 10\%$. Transforming the named $b_{s,N}$ values to numbers: for SWE in 2015/16, we observe a likelihood of $p = 0.9$ that all extrapolated 20 m by 20 m pixels range from 275–311 kg/m$^2$. In May 2017, extrapolated SWE values for an area of 4 km$^2$ are at 266–326 kg/m$^2$ with a likelihood of $p > 0.9$.

    The scaled cumulative probability distributions in Figure 6d demonstrate how representative a randomly located snow pit would be for the entire surveyed area. We analyzed both, the sample resolution corrected radar data (dotted lines) and the

filtered data (solid lines - see Section 2.2). The filtered data in Figure 6d indicate that SWE measured in a snow pit anywhere within the radar survey (black lines Figures 6a, b) would be within the error margins of $\pm 10\%$ from the mean of the entire kriged area with a high probability ($p = 0.99$ for winter accumulation in 2015/16 and $p = 0.91$ for 2016/17). The unfiltered



data, which represent not averaging snow depth around the snow pits, show that a point measurement would be within $\pm 10\%$ of the mean with $p = 0.89$ in 2015/16 and $p = 0.77$ in 2016/17. Such values demonstrate that SWE data derived from a snow

pit and not averaging snow depth around the pit will decrease the area-wide representativeness at Dye-2.

It is hard to explain the very low SWE variability in May 2016 at Dye-2. In theory, low wind speeds could lead to the absence of snow dunes and sastrugis and reduce the spatial heterogeneity of SWE. However, the recorded wind data do not confirm below average wind for this respective winter season. Determined statistics for wind speeds per winter season (01 Oct. – 01 May) at Dye-2 are very consistent over the last six years (2011/12 – 2016/17). We can only speculate that a snow fall event five

days prior to the radar measurements in May 2016 caused the low spatial variability in SWE.

At KAN-U, the transect lengths and area coverage differed greatly between May 2013 and May 2017 (Figures 7a, b). The 2013 survey covered an area of 1 km$^2$ with a transect length of more than 15 km. In 2017, our radar surveys were approximately 11 km in length resulting in extrapolated area coverage of 1.8 km$^2$. The average winter accumulation for the entire area are at 272 kg/m$^2$ ($\sigma = 20$ kg/m$^2$) in 2013 and 253 kg/m$^2$ ($\sigma = 19$ kg/m$^2$) in 2017.

The box plots in Figure 7c demonstrate a more variable data distribution at KAN-U than at the other two sites. The IQR for extrapolated SWE in 2012/13 is at -6 to +5% around the arithmetic mean. In 2016/17, the IQR decreases to $\pm 4\%$ around the mean. For both years, the whiskers reach outside the error margins of $\pm 10\%$ and, consequently, indicate less than $p = 0.9$ of data being within the error margins at KAN-U ($p = 0.82$ for 2012/13 and 2016/17). SWE data of 2016/17 has a higher skewness of 0.37 in comparison to 2012/13 (skewness of 0.17). Similar to the recorded radar data (see Figure 3), the upper

quartile in SWE is right shifted towards higher values. This is due to the homogeneous peak in accumulation at the northeastern corner of the grid (Figure 7b). Here, we measured above-average SWE, which consequently led to above-average interpolated values. The larger spatial heterogeneity in accumulation at KAN-U than at Dye-2 and Swiss Camp results in snow pits being slightly less representative of the surrounding area; only 80% of the respective May pit locations would provide area-wide SWE values being within a 10% error (for both accumulation seasons). Again, if snow depth is not averaged around pit locations the

likelihood of representing area-wide SWE decreases to $p = 0.68$ (2012/13) and $p = 0.64$ (2016/17).

Not all radar transect patterns collected are ideal for the applied geostatistical analyses. The distances between radar lines at Dye-2 and KAN-U in May 2017 are too large to allow interpolation between the lines. We had limited time available for radar surveys, and we chose to focus on surveying larger areas (up to 20 km$^2$) instead of on surveying on dense grids. The results presented in Figures 6b and 7a give us confidence that the data gaps do not include major dips or peaks in snow accumulation

because no such inhomogeneities exist within the areas of good spatial coverage.

The above results imply that one point measurement of SWE (snow pit, upGPR value, neutron probe, etc.) is representative for an area of roughly 4x4 km$^2$ at Dye-2 with a probability of $p \geq 0.9$ and an uncertainty of $\pm 10\%$ in case snow depth is averaged. For KAN-U, the spatial variability is slightly higher and, consequently, there is less certainty about how well a single measurement represents the surrounding area. However, we consider a probability of $p \geq 0.8$ with uncertainty of $\pm 10\%$ as

a resilient estimate. We recommend to combine a larger number of snow-depth probings within an area of 20 m by 20 m around the pits to increase the spatial representativeness of single snow accumulation measurement. Determining areal SWE in a single snow pit and without determining larger scale averages in snow depth results in a decreased probability of measuring





the area-wide average. Snow density varies only little within an area of several square kilometers, but snow depth can easily

vary by 10 cm or more on scales of several meters. The wind-induced surface roughness in particular has to be accounted for

to provide spatially-representative SWE values.

Averaging radar traces within 1 m radius results in a pit size of roughly 3 m². This is slightly too big for conventional pits

with on average 1 m snow depth. However, the search radius is related to the horizontal data resolution of the radar traces and

had to be further increased for the KAN-U site in 2012/13.

### 3.3 Accumulation pattern persistence

For KAN-U and Dye-2, we can analyze radar transects for two winter accumulation seasons. However, multi-year intersecting

radar transects and, hence, spatially-consistent area-wide SWE estimates for both sites are reduced. At KAN-U only 0.16 km²

of area was covered during both radar acquisitions. The intersecting area at Dye-2 comprises roughly 1.7 km² and, conse-

quently, more reliable conclusions can be drawn about accumulation-pattern persistence. At Dye-2, we observe a slight trend

in the north - south direction for both accumulation seasons at Dye-2 (Figure 6a and b). While the most southerly parts of the

transect show above area-wide average SWE values, the northern fringes are below the arithmetic mean of the area in SWE.

However, for both years the trends (in north to south direction) are statistically non-significant and very low at 5 kg/m² per

1 km for 2015/16 and 8 kg/m² per 1 km for 2016/17. The respective coefficients of determination of accumulation with latitude

are very low as well ($R^2 = 0.15$ - 2015/16 and $R^2 = 0.25$ - 2016/17). The parallel stripes, mainly visible in Figure 6b for the

southern parts are certainly artifacts provoked by the grid design and the applied kriging. Local maximums in regular distances

(150 – 220 m) occur along the transect line, however, the spatial extrapolation of these features is impossible due to the applied

radar grid.

To quantitatively assess agreement in accumulation patterns, we scaled accumulation data sets by their respective arithmetic

means, i.e. at Dye-2 divided the 2016 data by the 2017 data; the cumulative data distribution of the quotients is presented in

Figure 8. A constant area-wide quotient of 1 would imply that the scaled accumulation patterns are exactly equal. For Dye-2,

the probability of data being equally distributed in May 2016 and 2017 with a given uncertainty of ±10% is $p \geq 0.95$, meaning

all intersecting locations of the accumulation pattern in two consecutive years at Dye-2 are very similar. We did the same for

KAN-U, and the probability of similarity in accumulation pattern decreases to roughly $p = 0.85$; we note that this is for a small

area, and the radar observations were not made in consecutive years.

### 3.4 Temporal changes in accumulation at Dye-2

During snow pit measurements in May 2016, we placed target reflectors at the EMS 2015 surface in each pit. These targets

appear as hyperbolas in the radar data and make it possible to unambiguously identify that specific EMS for every subsequent

radar campaign. We identified several targets in the May 2017 radar data. Hence, it is possible to analyze changes in SWE

that occurred between May 2016 (the last radar campaign) and the end of 2016 melt season (i.e. the start of the 2016/2017

accumulation season). However, these analyses are only possible for intersecting areas of subsequent radar campaigns, which

is 1.7 km² at Dye-2. The area for which both the summer 2015 and summer 2016 IRHs could be clearly identified decreases to





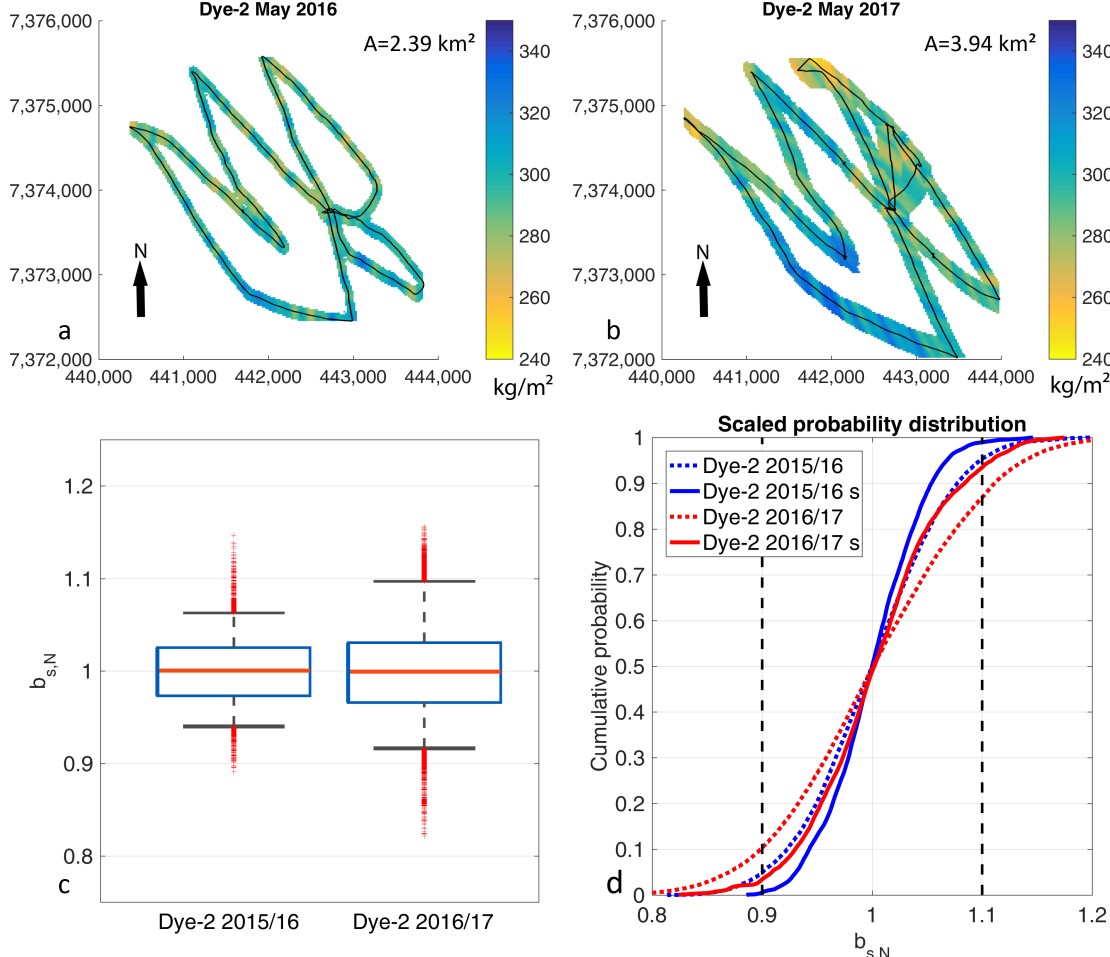

**Figure 6.** (a),(b) Kriging results for the radar transect at Dye-2 for snow accumulation. The black lines show the recorded radar traces and the black arrows indicate geographic north. $A$ is the total area covered by extrapolation. (c) Box plots displaying scaled data distribution ($b_{s,N}$) of kriged outputs with the red horizontal line showing the median, the blue box framing the interquartile range and the whiskers displaying the 5% and 95% percentiles. Outliers are shown as red crosses. (d) Scaled cumulative probability distribution of radar-derived SWE within any subset with 1 m radius of the GPR transects. The dashed vertical lines represent the determined error margins of $\pm 10\%$. Compared are filtered (solid lines and letter s) with non filtered (dotted lines) radar transects. All map coordinates are given in UTM with datum WGS1984.

0.76 km$^2$. Ice movement contributes to uncertainties as well. Identical locations in May 2016 and May 2017 do not represent identical snow and firn layers, since we observed horizontal ice movement of 25 m at the upGPR location.

Instead of snow pit data, we used a firn core (drilled in May 2017) to find the density of the layer between the 2015 and 2016 IRHs and to derive accumulation from TWT data as described in Equations 1 and 2. The firn between the 2015 IRH and the 2016 IRH is the net accumulation (accumulation minus meltwater percolation) between EMS 2015 and EMS 2016 ($b_{s,net}$),

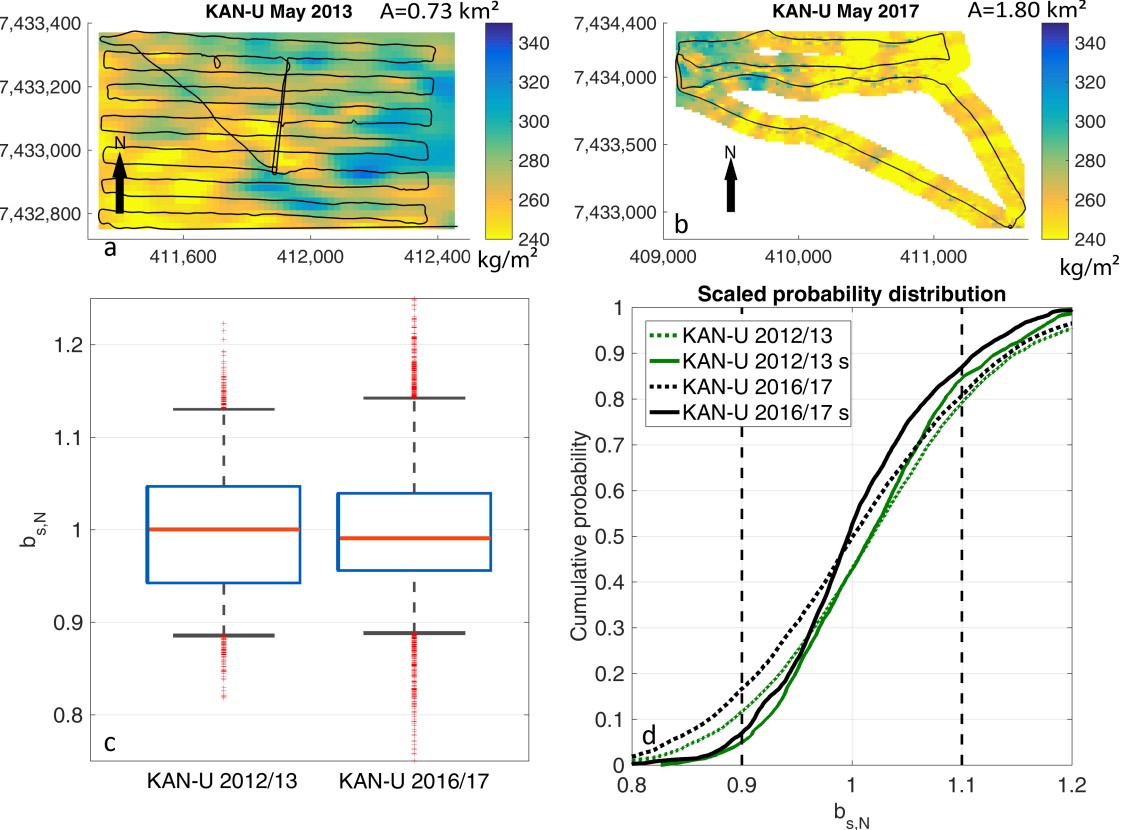

**Figure 7.** (a),(b) Kriging results for the radar transect at KAN-U for snow accumulation. The black lines show the recorded radar traces and the black arrows indicate geographic north. $A$ is the total area covered by extrapolation. (c) Box plots displaying scaled data distribution ($b_{s,N}$) of kriged outputs with the red horizontal line showing the median, the blue box framing the interquartile range and the whiskers displaying the 5% and 95% percentiles. Outliers are shown as red crosses. (d) Scaled cumulative probability distribution of radar-derived SWE within any subset with 1 m radius (3 m radius for 2012/13) of the GPR transects. The dashed vertical lines represent the determined error margins of $\pm 10\%$. Compared are filtered (solid lines and letter s) with non filtered (dotted lines) radar transects. All map coordinates are given in UTM with datum WGS1984.

whereas the radar data collected in May 2016 is the winter accumulation ($b_s$), from EMS 2015 to May 2016. The changes that occurred over summer 2016 $\Delta b_s = b_{s,net} - b_s$ are the subtraction of the winter accumulation from the net accumulation for area intersections. The mean $\overline{Delta b_s}$ for the intersecting transect areas (Figure 9a) for summer 2016 is 51 kg/m$^2$ with a standard deviation of 21 kg/m$^2$. Figure 9 documents the wide range in the data distribution. The negative values in Figure 9a occur only for six pixels and are likely artifacts.

Data distribution for $b_{s,net}$ is shown in Figure 9b as scaled values. Here, the distribution is less narrow than the winter accumulation in May 2016 (Figure 6d, blue line). Within the error margins, SWE decreases from $p = 0.99$ to $p = 0.88$ after one summer season. During summer 2016, melt caused a seasonal mass flux of 56 kg/m$^2$ into firn below EMS 2015 at the

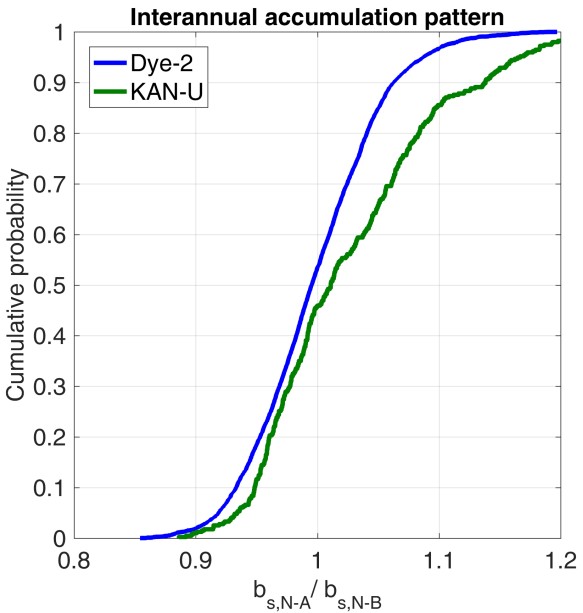

**Figure 8.** Interannual accumulation pattern comparison for intersecting transects at Dye-2 (blue line) and KAN-U (green line). Compared are scaled accumulation values ($b_{s,N}$) for both sites and divided by the latter year. At Dye-2 (KAN-U) $b_{s,N-A}$ corresponds to scaled SWE for winter accumulation 2015/16 (2012/13) and $b_{s,N-B}$ to 2016/17 (2016/17).

upGPR site (Heilig et al., 2018). It is very likely that the seasonal mass flux is not homogeneous over the investigated area.

This increased variability is in part due to mismatches in co-locating transects due to the ice movement. However, the mean change in SWE during summer 2016 corresponds almost exactly with observations derived from the upGPR (Heilig et al., 2018), which is 50.9 kg/m$^2$ from 01 May 2016 until the end of the melting period. This may be a coincidence or a confirmation of the benefits of upGPR, which averages a surface area of up to 10 m$^2$ compared to 1–3 m$^2$ area of a snow pit.

We cannot identify trends in SWE over the summer melt in 2016 associated with elevation; there are large differences within

the same elevation band (Figure 9a). This implies that (i) no lateral redistribution of mass can be observed at Dye-2 during snow and firn melt and (ii) that melt and seasonal mass fluxes are much more inhomogeneous than accumulation distribution. These conclusions support the assumption made by Heilig et al. (2018) that in the current climate there is no systematic lateral mass redistribution during the melt season at Dye-2.

We also measured SWE in snow pits near the upGPR at Dye-2 in May 2018 and 2019. Although the accumulations measured

in May 2016 and May 2017 were similar, the 2018 and 2019 data were much more varied (Table 4). In 2018 it was more than 20% higher than in the previous two accumulation seasons. The SWE measured in May 2019 was the lowest of the four years by a significant margin: 40% than the previous season and 23% lower than the next-lowest season (2017). This interannual accumulation variability is larger than the ±10% uncertainty in how well a SWE point measurement can be derived from radar data and usually represents the surrounding area. In agreement with Koenig et al. (2016), we conclude that annual or more



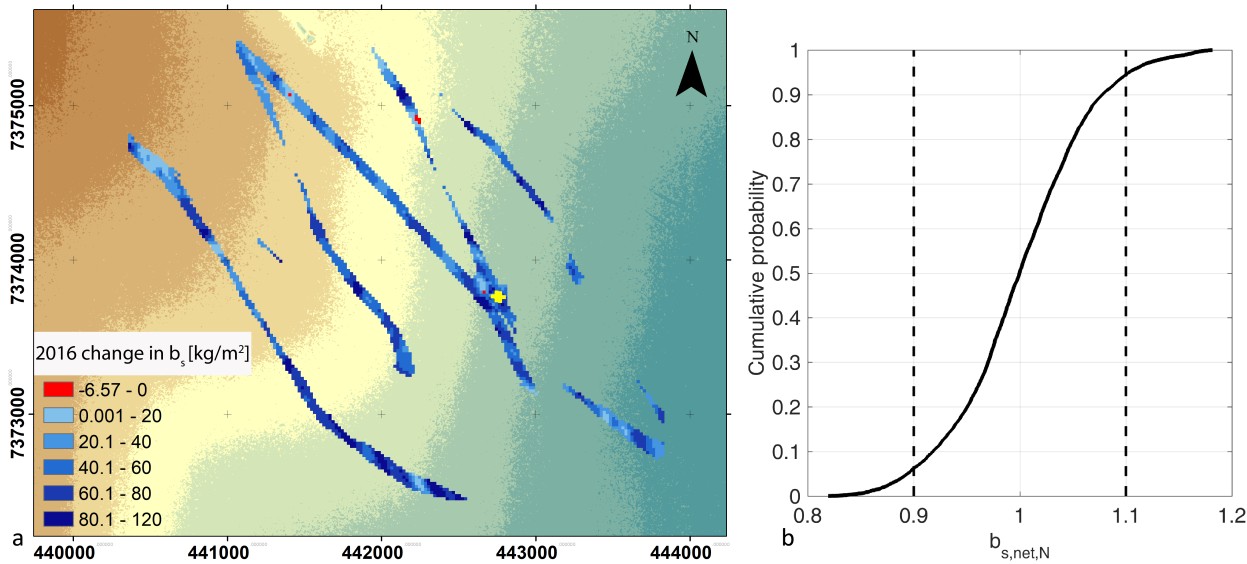

**Figure 9.** (a) Determined changes in SWE ($b_s$) from May 2016 until September 2016. The yellow crosses show locations of the upGPR. The background displays a 2 m DEM (Porter et al., 2018) with 5 m contour color coding starting at 2120 m a.s.l. (brown color) and reaching 2165 m a.s.l. (blue color). The coordinates are given in UTM with datum WGS1984. (b) Cumulative probability distribution of the scaled net accumulation ($b_{s,net,N}$) of the layer between end-of-melt-season 2015 and end-of-melt-season 2016 layers.

**Table 4.** Winter snow accumulation ($b_s$) and snow density ($\rho_s$) measured in spring snow pits for Dye-2 and KAN-U compared with determined area wide arithmetic means.

| Location date | $b_s$ snow pit [kg/m$^2$] | $\overline{b_s}$ kriged area [kg/m$^2$] | $\overline{\rho_s}$ [kg/m$^3$] |
|---|---|---|---|
| Dye-2 May 2016 | 313 | 293 | 320 |
| Dye-2 May 2017 | 294 | 296 | 334 |
| Dye-2 May 2018 | 372 | — | 361 |
| Dye-2 May 2019 | 225 | — | 364 |
| KAN-U May 2013 | 319 | 271 | 358 |
| KAN-U April 2017 | 246 | 252 | 316 |

frequent density and SWE observations are necessary to estimate mean accumulation rates per region correctly. When snow depth is measured and averaged over an area of roughly 20x20 m$^2$, the value provides a reliable estimate of accumulation on regional scales of 1–20 km$^2$. Such data can be used for airborne radar campaigns and for validation of RCM simulations.



## 4 Conclusions

This study investigated how representative single point observations of SWE, such as snow pits, are of the surrounding 400 m$^2$
to 4 km$^2$ large region. We used GPR to track IRHs created by summer melt surfaces along transects at three sites on the south-
western GrIS over the course of several field seasons. We derived maps of snow accumulation variability and compared them
to snow pit and upGPR measurements. We found an uncertainty in radar-derived accumulation of 7–8%, which results from
neglecting density variations along the radar transect and from applying a smoothing algorithm to minimize surface variability
and layer-picking errors. In addition, we analyzed the persistence of spatial patterns in accumulation over consecutive years
and the influence of melt on an annual firn layer.

We found that point measurements such as snow pits represent the average SWE well over the study areas at all three sites.
A randomly selected snow pit location at any of the three sites would provide an area-representative SWE (i.e. within ±10% of
the areal mean) with a probability of $p = 0.8$ (KAN-U May 2013) to $p > 0.95$ (Swiss Camp May 2015 and Dye-2 May 2016).
These likelihoods are independent of the size of investigated areas. However, not measuring and averaging snow depth over an
area of at least 20 m x 20 m decreases the probability of hitting arithmetic means by at least 10%. Snow-density variability is
usually below ±5% on regional scales, while snow depth can vary significantly because of surface features such as dunes and
sastrugi with various wavelengths ranging from submeters to 20–30 m.

Our results suggest that there is interannual persistence of accumulation patterns at least at Dye-2. However, the data only
span two consecutive accumulation seasons that were very similar in average density and accumulation. As such, we cannot
confirm whether such persistence might be observed in seasons with significantly more or less accumulation or at different
sites; this is a topic for future work.

We also investigated the mass change that an accumulation layer (end of melt season to May) undergoes during the summer
melt season using the GPR-transect data and continuous melt and accumulation observations from upGPR. We conclude that
temporal changes in firn layer mass detected by the upGPR are representative of larger ($\sim$1 km$^2$) areas at Dye-2. We did
not detect any patterns in summer melt along flowlines, suggesting that lateral meltwater flow at Dye-2 is not significantly
redistributing mass. However, this could change with future warming in Greenland, which would influence data interpretation
significantly of point measurements (AWS data, snow pits) and regional predictions by RCM and remote sensing.

This study aims to close the gap between point observations of SWE, which are meter scale, and remote-sensing data and
RCMs, which have pixel sizes of $\sim 1 - 20$ km. We have shown that snow accumulation in the regions surrounding the three
sites of the southwestern GrIS can be estimated well by point measurements as long as the snow depth is not influenced
by surface roughness. To minimize such roughness effects, it is essential to determine the average snow depth over an area
of several square meters. Ideally, snow-depth determinations - either directly via probings or derived from GPR transects-
comprise spacings in between single points smaller than the characteristic length of the features and have an extent larger
than the wavelengths of the features. Our data suggest that snow density does not vary greatly over km scales, and as such a
single density measurement with numerous probed depths can suffice. Because interannual variability in accumulation can be
significant, field measurements are essential for validating RCM predictions and remote sensing products.





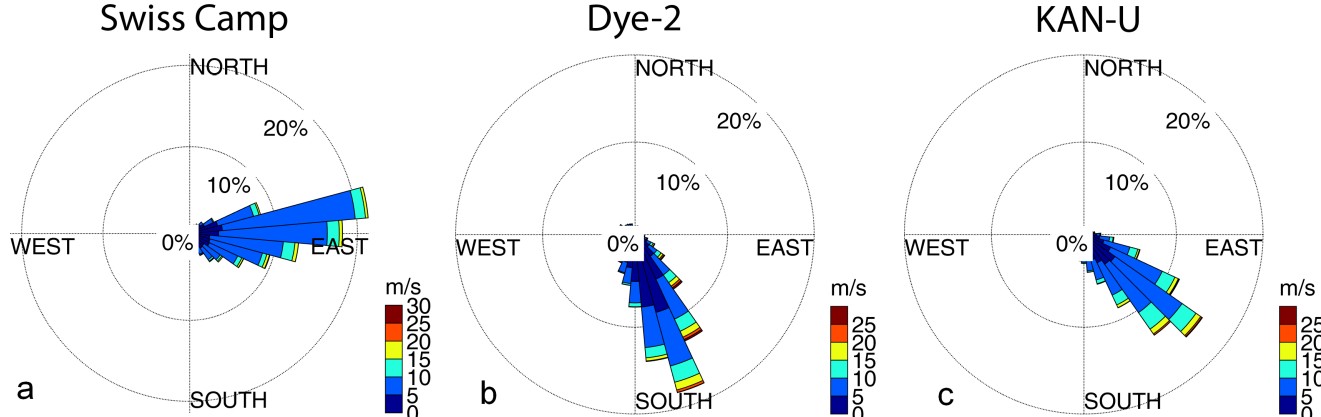

**Figure A1.** Prevailing wind distribution at Swiss Camp (a), Dye-2 (b) and KAN-U (c).

*Data availability.* All GPR transects will become available on public data bases within the end of 2019. If needed earlier, the data are available from the lead author upon request. All other parameters are presented within this manuscript.



*Competing interests.* The authors declare that they have no conflict of interest

*Acknowledgements.* A. Heilig was supported by DFG grant (HE 7501/1-1). M. MacFerrin and C.M. Stevens were supported by the National Aeronautics and Space Administration (NASA) grant NNX15AC62G. We highly acknowledge support in logistics and preparation of the field campaigns from K. Young and staff from Polar Field Services. We appreciate the support by Christoph Mayer and the Geodesy and Glaciology group at the Bavarian Academy of Sciences and Humanities who provided the radar equipment in 2017 and had numerous suggestion and recommendations improving this work. We thank B. Gerling, L. Gambal, S. Samimi, T. Snow and S. Marshall for their
assistance during field seasons.



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
