# Peer review of "Relating regional and point measurements of accumulation in southwest Greenland"

_The Cryosphere, 2019_

## Referee Comment (RC1) · Lynn Montgomery (Referee) · 16 Oct 2019

This paper investigates spatial variability in accumulation and density between regional scales and point measurements. Three locations in Southwestern Greenland are examined, two in the percolation zone and one at the equilibrium line, and gather observations including density measurements and ground penetrating radar transects. The manuscript attempts to answer four questions: error of internal horizon tracking data from GPR, how well a single point measurement of accumulation represents the larger area, analyzing the inter-annual variability of accumulation over several seasons, is melt and percolation affecting accumulation pattern? The resulting findings are that

there is a 7-8% error on IRH tracking mainly due to density uncertainties which is less than that of other methods like deriving accumulation from airborne radar as well as that point observations at all three sites do represent the average annual accumulation well over larger areas. Additionally, there is some inter-annual consistency for the DYE-2 site, though it was only sampled two years continuously and other sites are inconclusive. The final result is that currently melt is not effecting accumulation patterns.

This is clearly an important study which adds quite a bit of knowledge to our community about the spatial variability accumulation and density in Southwest Greenland. The results that point measurements represent larger areas is very impactful. Overall, the science is sound and credible.

However, my main point of concern is that the manuscript is extremely technical and difficult to follow at some points with concepts that require prior knowledge. It may discourage readers who are not fully comfortable with more in depth details of radar and some of the geostatistical methods. Clarification on several topics, detailed below, is needed for this to become a more readable paper.

Specific Comments: 1) The title is broad and a bit misleading. Three sites are examined in Southwest Greenland, however they are not representative of that entire area (as you state you can look at km wide results from this study). Along with this, the temporal aspect is questionable since there were only two consecutive years compared at Dye-2. The title should be narrowed to better represent what is being shown in the paper – i.e. "Relating regional and point measurements of accumulation in Southwest Greenland"

2) Major Questions (L70-78) – The manuscript attempts to answer quite a few questions (4 stated in the end of the introduction). Question (ii) is your main gap for this study, we do not know how representative point measurements are on a spatial scale and this could be the main focus of the opening since the majority of the paper is about it. In the process, you determine internal reflection horizon error of radar measurements because that is necessary to see how accurate your measurements are, so (i) can be removed. Question 3 is important, though you only have two years of consecutive data at Dye-2 to work with, is this really a main research question of the manuscript or can it just be addressed in the text? Question 4 is unclear, and a sentence follows to attempt to clarify it, however, it should be able to stand on it's own. Are you trying to ask if meltwater percolation effects IRH layers?

3) Depending on the background of the reader, there is a lot of jargon in this article especially in the methods section. The manuscript should be generally self-contained and the reader should not have to dig too deep outside in other literature for concepts that are discussed. Specific topics in the paper that could use more clarification are the radar processing (L99-102), vertical sampling (section 2.2), and variograms/kriging (Section 2.3, Table 3). Even if just a few sentences are added as background that would be helpful, see more specific comments below. Additionally, accumulation and SWE are used interchangeably in the text and figures in the manuscript. Be consistent with your terminology and use one or the other after you define what it is. Using both may confuse the reader if they are not familiar with this area.

In-Line Suggested Changes:

L 11-13 Re-arrange sentence for clarity. "Randomly selected snowpits are . . . occurring with a probability of p = . . ."

L23-24 Can you move the citations to the end of the sentence? The placement interrupts the flow.

L27 Include Enderlin et al, 2014 reference (An improved mass budget for the Greenland ice sheet, https://doi.org/10.1002/2013GL059010) along with van den Broeke.

L28 "(with positive and negative sign)" – what are you referring to here? Needs to be clarified.

L29 "negative trends in SMBs", SMB should not be plural.

L29 "Most of the GrIS, accumulation is dominating factor.. negative trends related to surface melt and runoff" where are these positive and negative trends occurring? Clarify.

L30 Remove "Despite their importance for the GrIS mass balance"

L32 Snowfall can be measured by remote sensing through satellites (i.e. CloudSAT). I.e. Bennartz et al, 2019 (https://doi.org/10.5194/acp-19-8101-2019), etc.

L33 "in concert" use another phrase here, take out "dedicated".

L47 Remove "worked to" and change "link" to "linked".

L52 "Other recent studies at" to "Other recent studies on"

L56-58 "Still, quantification.." This is repeating the same point as earlier in the paragraph (L47) Probably only need to state this once even though it is an important point.

L61-69 This paragraph is a bit disjointed. It begins with surface melt affecting SMB to annual accumulation estimates and observations to validating RCMs to melt impacting firn layers and then stating that there is a gap in how melt impacts temporal changes in accumulation distribution. Needs better flow.

L72 Remove "of altitude".

L70-78 See comments in Major Questions.

L76 Clarify Question (iv) if it is kept here. It should be clear enough on its on that there should not be a "In other words" after.

L89, L83, L94 Remove coordinates and elevations from text and include this in table 1. It is very distracting.

L99-102 Can this small section on radar units be combined with the paragraph above? Or can it be taken out and part of the table with a radar unit column?

L104 Include a sentence or small clause about what and why dewow and bandpass

filters for those who are not spun up about radar terminology.

(L116?) Equation 2 - Define beta.

L127 – Could you include the depth of the bulk density that you took from the snow pits?

L129 – Why do you include NASA-SE and EKT? They do provide you with two more range values but they are not relevant for SW Greenland. These sites are not brought up again later for any other analysis so could they be removed?

L133 – "For all three sites", similar to comment above, you are talking about five sites in this section but now only reference three in SW Greenland.

L138 – Is vertical sampling related to the frequency of the radar? If so, state this. Also, what is an example of small scale surface roughness? Are these not wind features?

L149 – Change "picked consistently" to "consistently picked"

L172 – Need an explanation of variograms prior to using it consistently throughout the next section.

L174 – Add a comma after "First"

L175 – Clarify "there are no gaps in accumulation in between", are there no gaps in the radar transmission of the accumulation?

L186 – "Despite the trend removal, anisotropy of the covariance...", unclear on what this means?

L187-193 – Using variograms consistently now, the term or concept needs to be explained prior for readers unfamiliar.

L197-198 – Define bs and bn in the sentence before the equation. They are stated but adding in the variables adds another layer of clarification.

L198 – "In the following" – what is this referring to? The following figure(s)?

L199 – Re-arrange this sentence. "Using the recorded radar traces, it is determined whether any randomly located..."

L208 – Back to the "Major Questions" point brought up above, the step to assess errors associated with TWT is necessary for your main question (ii) of the paper. This is stated as the first sentence. Is it necessary for this to be a major question in the opening since this is already a part of answering your other question? Clearly, this is a major result and should be discussed (as it is in the paper) but it is not necessarily the focus as the other question(s) are.

L210 – Is "accumulation pattern persistence" the same thing as inter-annual variability? The analysis is how accumulation is changing over space and time.

L211 – The wording of "whether seasonal changes in accumulation due to melt and liquid water percolation have major effects on accumulation pattern" is confusing. How would there be seasonal changes in accumulation due to melt? What is meant by accumulation pattern? How accumulation would change spatially due to melt? Is the question about how meltwater influences thickness of the layer? Please clarify this.

L222-223 – how deep were these snow pits?

L221 – Is it five locations in SW Greenland? The NASA-SE site is in SE Greenland, though the EKT site could be considered to be in SW Greenland.

L247 – Is "8-10m" the scale of the wind generated surface features causing minimums?

L252 – Earlier the "SWE" is referenced as "scaled accumulation", bs.N. (Section 2.3) Can you reference back to this for clarity as the variable?

L293-295 – "which represents not averaging snow depth around the snow pit". This is unclear, why would the area around the snow pit be averaged in?'

L324 – "However..." does this refer to KAN-U? Can you combine this with the previous sentence for clarity?

Section 3.3 – Frankly, could take out the KAN-U comparison with such a small area overlapped and not having consecutive years of data, there is no real major conclusions to be drawn here and it is not brought up again in the paper.

L349 – example here to explain the figure is great for clarity.

L363 – Will using a firn core instead of a density pit induce any further uncertainty?

L368 – Can you use a Delta symbol?

L370 – Artifacts in the sense that the GPR data from the winter accumulation was greater than the net accumulation?

L375 – If the ice movement is known from the upGPR site, can it be corrected for?

Section 4 – Conclusions – This very nicely ties up the study concisely and answers the questions put forth in the opening. If some questions are taken out, needs to be revised.

Figures and Tables Comments:

For all figures: make sure if it is a multi-paneled plot, that the (a)/(b)/(c) are either inside or outside of the figures consistently. i.e. figure 4 a is outside the box and b is inside the box.

Table 1 – Move KAN-U April 2017 before Dye-2 May 2017 if table is supposed to be chronological. Include coordinates of sites and elevations in table. Possibly include radar units in table as well?

Figure 1 – Is OK how it is currently, but a little unclear - could take off elevation markings and put a color bar labeling elevations to see if it made the graphic a little less cluttered. Difficult to see the 500m marking near the coast. Not sure if the 1000, 1500m elevations are labelled at all?

Table 2 – See comments from above, take out NASA-SE and EKT if they are not

relevant for the rest of the study. Possible include depth of the pit? For the density ranges, could you use something other than "-" in the range column? It could confuse the reader if they did not scan the text for what the range meant.

Figure 3 – Recommend using colors other than red and green stacked on one another for colorblind purposes. Could use a dashed line or a thicker line of another color.

Figure 4 – Similar comment to above, try a different color than red and green since the lines are close to one another. Could use transparency to see the standard normal distribution behind.

Figure 4/ Figure 5 – Can these be combined into a 3 panel plot since you're talking about the same area?

Table 3 – Never described what the major and minor axis are in the text for variograms. Does the mean prediction error need to be in the table if they are all 0 (expect the first value of 0.01)?

Figure 9 – Need colo rbar for the contour on 9a, difficult to see two locations of upGPR.

Figure 4,6,7,9 – A personal preference is to have coordinates in lat/lon instead of UTM. If the scales do not allow though, especially for an area like Swiss camp, that is fine because it is on a few km scale.

Other comments:

The word "very" is used quite a bit throughout the manuscript as a qualifier and those instances can be removed the majority of the time. Using it does not add to the meaning of the sentences.
* * *

---

## Referee Comment (RC2) · Anonymous Referee #2 · 30 Oct 2019

OVERVIEW

This paper tries to answer the question of how representative point measurements of snow accumulation are for the larger regional scale. This subject is important, urgently needs attention, and this paper fills a void in our knowledge on the connection between the observation scale and the (regional) climate modelling scale.

Scientifically, the paper is solid, and I have few methodological remarks. In terms of presentation however, I have quite a few remarks. Changing a word or sentence here or there won't fix the fact that the paper is quite tough to read.

GENERAL

[Figure]

- Various terms are used interchangably, without a proper definition. Snow accumulation, SMB, SWE, snowfall, snow depth. Please have a critical look at the terminology, simplify, and make uniform.

- I had to dig quite deep in my memory to connect the dots between variograms, nuggets, space-invariance, isotropy and stationarity. Would it be feasible to ease the text in section 2.3?

- The abstract is particularly awkward in grammar and style, as if it was the last part that was written and not checked before submission. I'll give three example sentences and how to make this readable:

1. "Density variations per site for snow pits within distances of up to 1 km are found to be consistently within 5%."

–>

"Within one km of each site, the density consistently varied within 5%."

2. "It occurs that with a probability of p = 0.8 (KAN-U) to p > 0.95 (Swiss Camp and Dye-2), randomly selected snow pits are representative in snow accumulation for entire regions with an offset of 10% from arithmetic means."

–>

"We found that snow accumulation from a randomly selected snow pit is very likely representative of the regional scale (p = 0.8 for a value within 10% of the regional mean for KAN-U, and p > 0.95 for Swiss Camp and Dye-2)."

3. "Interannual accumulation pattern at Dye-2 are very persistent for two subsequent accumulation seasons with similarity probabilities of p > 0.95, if again an error of 10% is included."

–>

"At Dye-2, we found that the spatial pattern of snow accumulation was very similar for two consecutive years."

So please revise the manuscript text and simplify, simplify and simplify.

- In several parts, you claim that snow accumulation should be established for an area of at least 20 x 20 m. This seems a very important implication for future field work. However, I miss the quantitative underpinning of these numbers. Why not 10 x 10, or 25 x 25 m? And how should this be done if no GPR is available? This is such a crucial part of the manuscript that I expect some more discussion of the implications on field practice.

- The title is inappropriate. My suggestion would be: "Representation of point measurements for regional-scale snow accumulation in/on the southwestern Greenland Ice Sheet."

- Throughout the paper, you seem to use rho_s mostly as a bulk parameter: a mean over a certain depth. Can you more clearly distinguish between the actual snow density and this vertically integrated bulk density, and define the bulk density clearly?

SPECIFIC REMARKS

- Title: "South-Western". I looked it up but this should be either "southwestern Greenland" or "Southwest Greenland".

- L1: significant changes. In what?

- L3: remove the sentence "Data sources ... coverage"

- L3: remove "at least"

- L11: "per regional scale"?

- L11: "to analyze for". To analyze is a transitive verb. Suggestion "we investigate". This recurs frequently in the text (e.g. P1L17)

- L18: "cannot be evidenced" -> "At Dye-2, we found no evidence of..."

- L27: SMB related processes -> SMB

- L35: "certainly contribute" -> "may also contribute"

- L52: remove "at the GrIS"

- L57: point -> location

- L70: suggest "To improve our understanding of the representativeness of ..."

- L71: "in area for two sites" -> " in areas around two sites"

- L72: I have once been taught that one paper answers one major question. Your one major question is about representativeness of point measurements. All other questions are hurdles that you come across while answering that question. My suggestions would be to rephrase, and to formulate L70-80 such that you introduce the different steps needed to answer your "major question" with associated sections.

- L72: "equilibrium line of altitude" -> "equilibrium line"

- L125: numerous -> several

- L144: range -> ranges

- L146: constantly -> always

- L172: more constraint -> better constrained

- L181: remove "a realm of"

- L197: Do not start a sentence with a mathematical symbol.

- L196 - 206: past and present tense are used here interchangably. Please unify.

- L217: snow depths -> snow depth

- L217: why call this ice volume fraction? Suggestion to simplify these sentences:

"We investigate the error that we introduce by assuming a single bulk density in the conversion from TWT to snow depth for an entire GPR transect. For that, we use a collection of snow pits, several from each of five locations, that were collected in a period of three years (table 2)."

- L243: above average -> above-average (idem below-average)

- L261: equals to -> equals

- Figure 4: consider inverting the color scale. Blue = low accumulation, yellow is high accumulation

- L293: awkward construction. Rephrase

- L316: Not all of the collected radar transect patterns (grids?) ...

- L327: larger scale -> larger-scale

- L341: north to south direction -> north-to-south direction

- L347: this sentence is not complete

- L368: LaTeX error

- L387: 40% -> 40% lower

- L408: The conclusion about persistence is unsatisfactory, and you seem to be shifting goal posts in the manuscript. In the abstract you write that interannual accumulation patterns "are very persistent". In section 3.3, the 2016 and 2017 data are "very similar". Then in L408 you say that "results suggest persistence". I think you should refrain at all from inferring persistence based on two data points. It's ok to mention that the patterns were similar in both 2016 and 2017, but I don't think there is enough argument here to start discussing persistence.

---

## Editor Comment (EC1) · Joseph MacGregor (Editor) · 10 Nov 2019

Hi Dr. Heilig et al.,

Thanks again for your submission to The Cryosphere Discussions. I have examined the referees' comments and found them to be both constructive and uncommonly compatible. Both find value in the study of the compatibility of radar-mapped accumulation and snow pits, but they take issue with how the motivating questions are framed and the complexity of the language used to describe the results. I recommend that you hew closely to their comments in your response, and that your native English co-authors carefully review the manuscript to ensure your study is communicated as effectively as

possible.

Thanks,

Joe MacGregor

―――――――――――――――――――

---

## Author Comment (AC1) · 29 Nov 2019

We thank both referees and the associated editor for very constructive and helpful comments. There were several points raised by both referees that addressed similar or equivalent points. We listed the common points of criticism first before individual comments of each referee are considered separately. Minor changes such as typos have been incorporated in the MS without listing them here. In order to improve readability, comments by the respective referee are listed in italic, while responses and modifications in the MS are written in regular typesetting. Sentences and paragraphs being incorporated in the manuscript are listed in bold letters here and in the manuscript. To keep the manuscript up to date, we checked for recent publications and included some wherever appropriate. Within the introduction, we included Mottram et al. (2019) as another source for changes in mass loss processes and added Lewis et al. (2019) as another example of extensive ground-based radar campaigns. In addition, we exchanged the previously referenced Lewis et al. (2019) discussion paper in TCD to the now published Lewis et al. (2019) TC paper.

**Common points of criticism:**

- Both referees suggest to change the title of the manuscript. We decided to use the suggestion by Lynn Montgomery and changed the title to: **Relating regional and point measurements of accumulation in southwest Greenland.**

- Another point both referees criticize is the inconsistent/ interchangeable usage of SWE and snow accumulation within the manuscript. Surface mass balance (SMB) is solely used (and properly introduced, L27) within the introduction. Here, SMB is defined as …sum of snow accumulation and lateral redistribution by sublimation, wind and runoff…. This specifies the usage of the term "accumulation" and the importance of determining its spatial representativeness. In the revised manuscript, we consistently have changed the terminology to snow accumulation with symbol $b_s$ and units [kg/m$^2$].

- In addition, it has been suggested to simplify especially the section 2.3 dealing with spatial extrapolation. We now introduce terms such as variogram, nugget and anisotropy to facilitate readability of Section 2.3. Some radar terms are additionally explained as well.

- We modified the respective paragraphs in the introduction, which deal with objectives and scientific questions this work tries to answer. We fully agree that the main purpose of this manuscript is the relation of point measurements to regional accumulation. As stated by referee #1, the raised question (i) is a prerequisite to assess spatial representativeness and, hence, is removed from this listing. Since commonly applied in situ measurements of snow accumulations represent only a snapshot in time, it remains open whether accumulation patterns change with summer melt processes and are similar for two different winter accumulation season. We agree that the assessment of seasonal persistency cannot be properly determined with the available field data. However, since temporally continuous determinations of changes in accumulation are available and feasible in Greenland nowadays (upGPR, neutron probes), a relation of two consecutive years of data with point measurements is valuable and consequently is addressed in the results and discussion section. In addition, liquid water percolation has an effect on accumulation resulting in seasonal mass fluxes from the surface into deeper firn especially for

the investigated sites within the deep percolation zone of the Greenland Ice Sheet. We changed the respective paragraph to the following statement:

**The aim of this work is to relate point scales to regional scales of one to several square kilometers in area to improve our understanding of the representativeness of point measurements. For this purpose, we examine snow-pit and GPR data from two sites within the percolation zone of the GrIS and one site at the equilibrium line gathered over several field seasons. For each site, we investigate density variability between measurements from up to six snow pits within an area of 4 km$^2$ made in a single season, process radar transects of up to 25 km recorded in close proximity to those snow pits, and spatially extrapolate the radar-derived accumulation to estimate area-wide accumulation variability. For temporal comparisons, we use continuous observations of accumulation and melt recorded by upGPR \citep{Heilig2018}. Our results show that spatial representativeness of snow accumulation for a point measurement (snow pit) is high but values can be affected by local wind-induced surface roughness. We recommend to apply multiple snow depth measurements at the vicinity of the pits to better assess accumulation on regional scales.**

**Reply to referee #1 (Lynn Montgomery):**

We highly appreciate comments raised by the referee and present a point-to-point reply for all issues being listed. For an improved readability and to facilitate direct response, we sometimes subdivided comments into several paragraphs referring to similar issues

*Comments to the Author*
*Assessment*
*This is clearly an important study which adds quite a bit of knowledge to our community about the spatial variability accumulation and density in Southwest Greenland. The results that point measurements represent larger areas is very impactful. Overall, the science is sound and credible. However, my main point of concern is that the manuscript is extremely technical and difficult to follow at some points with concepts that require prior knowledge. It may discourage readers who are not fully comfortable with more in depth details of radar and some of the geostatistical methods. Clarification on several topics, detailed below, is needed for this to become a more readable paper.*

We appreciate the assessment by the reviewer, have facilitated readability and hope that it now meets expectations.

*Specific comments*
*The title is broad and a bit misleading. Three sites are examined in Southwest Greenland, however they are not representative of that entire area (as you state you can look at km wide results from this study). Along with this, the temporal aspect is questionable since there were only two consecutive years compared at Dye-2. The title should be narrowed to better represent what is being shown in the paper – i.e. "Relating regional and point measurements of accumulation in Southwest Greenland".*

See above – we changed the title accordingly.

*Major Questions (L70-78) – The manuscript attempts to answer quite a few questions (4 stated in the end of the introduction). Question (ii) is your main gap for this study, we do not know how representative point measurements are on a spatial scale and this could be the main focus of the opening since the majority of the paper is about it. In the process, you determine internal reflection horizon error of radar measurements because that is necessary to see how accurate your measurements are, so (i)can be removed. Question 3 is important, though you only have two years of consecutive data at Dye-2 to work with, is this really a main research question of the manuscript or can it just be addressed in the text?*

*Question 4 is unclear, and a sentence follows to attempt to clarify it, however, it should be able to stand on it's own. Are you trying to ask if meltwater percolation effects IRH layers?*

See above in the common introduction to our replies – we rephrased the entire paragraph to highlight the main purpose of the manuscript and to demonstrate that analysis of interannual similarities and lateral flow effects are necessary to increase the impact of this paper especially in terms of temporal generality. However, the stated questions are removed as suggested.

*Depending on the background of the reader, there is a lot of jargon in this article especially in the methods section. The manuscript should be generally self-contained and the reader should not have to dig too deep outside in other literature for concepts that are discussed. Specific topics in the paper that could use more clarification are the radar processing (L99-102), vertical sampling (section 2.2), and variograms/kriging (Section 2.3, Table 3). Even if just a few sentences are added as background that would be helpful, see more specific comments below.*

This point has been raised by referee #2 as well and, hence, is treated in the common section above. We rephrased and extended respective parts of the manuscript to address this criticism.

*Additionally, accumulation and SWE are used interchangeably in the text and figures in the manuscript. Be consistent with your terminology and use one or the other after you define what it is. Using both may confuse the reader if they are not familiar with this area.*

Agreed; see above, we changed SWE consistently to accumulation with symbol $b_s$.

*In-Line Suggested Changes:*
*L 11-13 Re-arrange sentence for clarity. "Randomly selected snowpits are...occurring with a probability of p =...".*

We have rephrased the abstract thoroughly as suggested by referee #2.

*L23-24 Can you move the citations to the end of the sentence? The placement interrupts the flow.*

Changed accordingly.

*L27 Include Enderlin et al, 2014 reference (An improved mass budget for the Greenland ice sheet, https://doi.org/10.1002/2013GL059010) along with van den Broeke.*

Changed accordingly

*L28 "(with positive and negative sign)" – what are you referring to here? Needs to be clarified.*

Exchanged the brackets statement to: "**Depending on the location, lateral redistribution can increase SMB as well as decrease it.**"

*L29 "negative trends in SMBs", SMB should not be plural.*

Changed accordingly.

*L29 "Most of the GrIS, accumulation is dominating factor.. negative trends related to surface melt and runoff" where are these positive and negative trends occurring? Clarify.*

Here, we refer to the GrIS in its entirety. We do not talk about specific regions within the ablation zone or areas where surface runoff or basal runoff are occurring. We included "**recent negative trends in SMB**" to clarify the temporal reference.

*L30 Remove "Despite their importance for the GrIS mass balance"*

Changed accordingly.

*L32 Snowfall can be measured by remote sensing through satellites (i.e. CloudSAT).I.e. Bennartz et al, 2019 (https://doi.org/10.5194/acp-19-8101-2019), etc.*
Here, we respectfully disagree. Bennartz et al. (2019) describe that "…CloudSAT provide ESTIMATES of snowfall in remote regions…". They present several sources of uncertainties and "…approaches to mitigate these adverse effects…". So we still keep the statement that snowfall cannot be measured but changed the phrase to: **This is because surface mass fluxes, such as snowfall and melt, cannot be measured by remote-sensing technology and derived estimates on snowfall can still have significant errors \citep{Bennartz2019}. Hence, predictions of SMB are usually obtained using scarce in situ measurements together with regional climate models (RCMs), which can introduce significant uncertainties \citep{Vernon2013} as well.**

*L33 "in concert" use another phrase here, take out "dedicated"*
Changed to: …**together**…; dedicated has been removed

*L47 Remove "worked to" and change "link" to "linked"*
Changed accordingly.

*L56-58 "Still, quantification.." This is repeating the same point as earlier in the paragraph (L47) Probably only need to state this once even though it is an important point.*
Changed to: "**Since quantification of spatial representativeness of single point measurements for the surrounding square kilometers has only been conducted for one point in western Greenland so far \citep{Dunse2008}, there is a need to explore uncertainties at local and regional scales.**". We consider this sentence as being valuable to highlight the motivation of this work.

*L61-69 This paragraph is a bit disjointed. It begins with surface melt affecting SMB to annual accumulation estimates and observations to validating RCMs to melt impacting firn layers and then stating that there is a gap in how melt impacts temporal changes in accumulation distribution. Needs better flow.*
Changed to: **Meltwater percolation can move mass from snow to the underlying firn (e.g., \citealp{Charalampidis2016,Humphrey2012,Heilig2018}) or even laterally along the surface slope \citep{Humphrey2012}. Hence, surface melt affects SMB (e.g., \citealp{Sasgen2012}) and accumulation \citep{Heilig2018}. However, it is unlikely that water percolation and mass redistribution are homogeneous over regional scales. Consequently, it is necessary to assess the impact of melt on temporal changes in accumulation distribution for the percolation zone of the GrIS.**

*L72 Remove "of altitude".*
Removed.

*L70-78 See comments in Major Questions.*
Answered as stated above.

*L76 Clarify Question (iv) if it is kept here. It should be clear enough on its on that there should not be a "In other words" after.*
Question has been removed.

*L89, L83, L94 Remove coordinates and elevations from text and include this in table 1. It is very distracting.*
Coordinates are now included in Table 1.

*L99-102 Can this small section on radar units be combined with the paragraph above? Or can it be taken out and part of the table with a radar unit column?*
Since the information in brackets on the respective coordinates of the measurement locations were considered as being distracting, we decided to keep the paragraph as is.

*L104 Include a sentence or small clause about what and why dewow and bandpass filters for those who are not spun up about radar terminology.*
Changed to: **All recorded radar traces were processed in a very similar way. In case first arrivals were delayed by more than approximately 2 ns, we started with a correction for the DC shift. Offsets in the zero line of each radar trace (wow) were corrected utilizing a dewow function and low (approximately below 0.5 times the center frequency) and high frequency noise (approx. above 1.5 times the center frequency) were cut by bandpass filters. We further applied background removals to minimize direct wave influences.**

*(L116?) Equation 2 - Define beta.*
We included: …**the exponent β=0.5 (related to a medium with random orientation at the micro scale), …**We apologize for this.

*L127 – Could you include the depth of the bulk density that you took from the snowpits?*
The bulk density is calculated over the entire snow column. We did not define samples of a specific depth as being representative of the bulk. As requested, we included snow depth values in Table 2 and included: **(see Table 2 for details).**

*L129 – Why do you include NASA-SE and EKT? They do provide you with two more range values but they are not relevant for SW Greenland. These sites are not brought up again later for any other analysis so could they be removed?*
We included description of the sites within the methodology and used the presented data for extension of the conclusions of regional spatial variability in snow density within the discussion section. **As these two sites are located within a distance of 45—60 km of the GrIS ice divide (W of the divide - EKT and E of the divide - NASA SE, see Figure 1), they extent our data analysis of spatial variability of $\rho_s$ to the dry-snow zone. The recorded pits at NASA SE provide data for a high accumulation site as well.**

*L133 – "For all three sites", similar to comment above, you are talking about five sites in this section but now only reference three in SW Greenland.*
Changed to: **For all three transect sites**…

*L138 – Is vertical sampling related to the frequency of the radar? If so, state this. Also, what is an example of small scale surface roughness? Are these not wind features?*
No, vertical sampling rates are related to the depth ranges selected (time window length of the radar acquisition) and the sampling frequency (how many samples are measured within the selected range). Since we intended to use the recorded data also for other purposes such as analyzing deeper firn stratigraphy, the selected range and sampling rates were a trade off in between vertical resolution and depth. Concerning the second question, you are right small scale surface roughness are mostly related to wind features as being introduced in the subsequent sentence.

*L149 – Change "picked consistently" to "consistently picked"*
Changed accordingly.

*L172 – Need an explanation of variograms prior to using it consistently throughout the next section.*
*L187-193 – Using variograms consistently now, the term or concept needs to be explained prior for readers unfamiliar.*
We extended the following sentence to introduce the term variogram: **\citet{Webster2007} state that sample size is directly related to the precision of variogram estimates, while variograms are used to estimate the variance of a parameter (here snow accumulation) at increasing intervals of distance in between measurements and in multiple directions.**

*L174 – Add a comma after "First"*
Changed accordingly.

*L175 – Clarify "there are no gaps in accumulation in between", are there no gaps in the radar transmission of the accumulation?*
We modified the phrase within the brackets to: …**(accumulation occurred everywhere within the area of interest, governed by local weather conditions).** However, the entire subsection changed significantly. **…**

*L186 – "Despite the trend removal, anisotropy of the covariance...", unclear on what this means?*
See above, the section was rephrased. The respective sentence reads now: **In addition, we found directional anisotropy of the covariance in all of the longer transects, which means that accumulation variation varies with direction.**

*L197-198 – Define bs and bn in the sentence before the equation. They are stated but adding in the variables adds another layer of clarification.*
Changed accordingly.

*L198 – "In the following" – what is this referring to? The following figure(s)?*
Changed to: **In Figures 4, 6 and 7,**

*L199 – Re-arrange this sentence. "Using the recorded radar traces, it is determined whether any randomly located..."*
Changed accordingly.

*L208 – Back to the "Major Questions" point brought up above, the step to assess errors associated with TWT is necessary for your main question (ii) of the paper. This is stated as the first sentence. Is it necessary for this to be a major question in the opening since this is already a part of answering your other question? Clearly, this is a major result and should be discussed (as it is in the paper) but it is not necessarily the focus as the other question(s) are.*
See above – the respective paragraph has been changed significantly and all listed questions are removed. As suggested, we focus on spatial representativeness whereas liquid water percolation as well as multiple radar acquisitions are supportive to assess representativeness and reach a broader impact as just singular point observations in time.

*L210 – Is "accumulation pattern persistence" the same thing as inter-annual variability? The analysis is how accumulation is changing over space and time.*
We have modified the language and removed term "accumulation pattern persistence". We now describe changes within the two consecutive accumulation season observations at Dye-2.

*L211 – The wording of "whether seasonal changes in accumulation due to melt and liquid water percolation have major effects on accumulation pattern" is confusing. How would there be seasonal changes in accumulation due to melt? What is meant by accumulation pattern? How accumulation would change spatially due to melt? Is the question about how meltwater influences thickness of the layer? Please clarify this.*

We clarified to: **Finally, we investigate how accumulation changes due to melt and liquid-water percolation.**

*L222-223 – how deep were these snow pits?*

We included a column in Table 2 with mean snow depths and included the following phrase: …distances between ranged from a few meters up to 1 km, **while snow depths ranged: from 0.83 m to 1.70 m**.

*L221 – Is it five locations in SW Greenland? The NASA-SE site is in SE Greenland, though the EKT site could be considered to be in SW Greenland.*

Changed to …**southern GrIS….** In addition, we included**: The inclusion of two more sites close of the southern Greenland ice divide extents the data set to a low accumulation site west of the ice divide (EKT: $\bar{b}_s$~300 kg/m$^2$) and a high accumulation sites east of the divide (NASA SE: $\bar{b}_s$~600 kg/m$^2$).**

*L247 – Is "8-10m" the scale of the wind generated surface features causing minimums?*

Changed to: **However, the observed minimums in b$_s$ along the south-north transect lines are at regular distances between 8—10 m and are likely the result of wind-generated surface features.**

*L252 – Earlier the "SWE" is referenced as "scaled accumulation", bs.N. (Section 2.3) Can you reference back to this for clarity as the variable?*

Modified to: **Figure 4b displays the scaled accumulation distribution (b$_{s,N}$) through box plots.**

*L293-295 – "which represents not averaging snow depth around the snow pit". This is unclear, why would the area around the snow pit be averaged in?'*

Changed to: **The unfiltered data, however, show a decreased representativeness with p=0.89 in 2015/16 and p=0.77 in 2016/17 for the same uncertainty range of ±10\%. Here snow depth is solely derived from the snow pit. Such values demonstrate that b$_s$ data derived simply from a snow pit without averaging snow depth for an area around the pit location will decrease the area-wide representativeness at Dye-2.**

*L324 – "However…" does this refer to KAN-U? Can you combine this with the previous sentence for clarity?*

We included: **However, we consider a probability of p≥0.8 with uncertainty of ±10\% for both study sites as a resilient estimate.**

*Section 3.3 – Frankly, could take out the KAN-U comparison with such a small area overlapped and not having consecutive years of data, there is no real major conclusions to be drawn here and it is not brought up again in the paper.*

Changed as suggested to:

**At KAN-U only 0.16 km$^2$ were covered during both radar acquisitions and, consequently, we do not investigate changes in accumulation for spring 2013 and 2017. For Dye-2, we recorded radar transects for two consecutive winter accumulation seasons. However, multi-year intersecting radar transects and, hence, spatially-consistent area-wide b$_s$ estimates are reduced. The intersecting area at Dye-2 comprises roughly 1.7 km$^2$. Here, we observe a slight trend in the north - south direction for both accumulation seasons (Figure 6a and b). While the most southerly parts of the transect show above**

**area-wide average $b_s$ values, the northern fringes are below the arithmetic mean of the area in $b_s$. However, for both years the trends (in north to south direction) are statistically non-significant and very low at 5 kg/m$^2$ per 1 km for 2015/16 and 8 kg/m$^2$ per 1 km for 2016/17. The respective coefficients of determination of accumulation with latitude are very low as well ($R^2$=0.15 - 2015/16 and $R^2$=0.25 - 2016/17).**
**The parallel stripes, mainly visible in Figure 6b for the southern parts, are certainly artifacts provoked by the grid design and the applied kriging. Local maximums in regular distances (150 – 220 m) occur along the transect line, however, the spatial extrapolation of these features is impossible due to the applied radar grid.**

**To quantitatively assess agreement in accumulation patterns, we used the respective normalized accumulation data and calculated the quotient.  The cumulative data distribution of the quotients is presented in Figure 8.  A constant area-wide quotient of 1 would imply that the normalized accumulation patterns are exactly equal. For Dye-2, the probability of data being equally distributed in May 2016 and 2017 with a given uncertainty of ±10\% is p≥0.95, meaning all intersecting locations of the accumulation pattern in two consecutive years at Dye-2 are similar.**

*L349 – example here to explain the figure is great for clarity.*
Thanks

*L363 – Will using a firn core instead of a density pit induce any further uncertainty?*
No, but it was impossible to dig down to the end-of-summer-horizon 2015 just using snow shovels. We did not have a chain saw with us for this field campaign and, hence, collected density data in a firn core. We do not consider that the firn core is providing more uncertainty but the method is different, which should be mentioned here.

*L368 – Can you use a Delta symbol?*
Corrected, we had a missing \ in the previous version. We apologize for this.

*L370 – Artifacts in the sense that the GPR data from the winter accumulation was greater than the net accumulation?*
Artifacts in the sense that the accumulation in September 2016 was higher than in May 2016. Due to the fact that summer melt 2016 was significantly above average in terms of area extent in surficial melt (see Heilig et al. 2018 for details), it is unlikely that for specific locations accumulation increased while the average decrease in $b_s$ is at 51 kg/m$^2$. Those artifacts most likely arise from singular outliers in kriged accumulation and are restricted to only six pixels. We included: ….**are likely artifacts due to kriging outliers and errors**… to clarify the sentence.

*L375 – If the ice movement is known from the upGPR site, can it be corrected for?*
We only have a rough location estimate from handheld GPS data. We do not consider such accuracies as adequate to correct all radar locations even though location uncertainties (5-10 m) are likely smaller than the annual ice movement (~25 m). However, accumulation values are extrapolated for 20 m by 20 m pixel sizes. It is debatable, whether co-locating GPR transects would decrease discrepancies of accumulation values from May 2016 to September 2016.

*Section 4 – Conclusions – This very nicely ties up the study concisely and answers the questions put forth in the opening. If some questions are taken out, needs to be revised.*
Although, we removed the questions from the introduction, we do not think that the conclusion section has to be changed significantly. The term "interannual persistence" was removed throughout the

manuscript. So the respective paragraph in the conclusions changed to: **Our results suggest that there is only little change of accumulation patterns at Dye-2 for spring 2016 and 2017. However, the data only span two consecutive accumulation seasons that were very similar in average density and accumulation. As such, we cannot confirm whether such persistence might be observed in seasons with significantly more or less accumulation or at different sites; this is a topic for future work.**

*Figures and Tables Comments: For all figures: make sure if it is a multi-paneled plot, that the (a)/(b)/(c) are either inside or outside of the figures consistently. i.e. figure 4 a is outside the box and b is inside the box.*
Has been changed as required.

*Table 1 – Move KAN-U April 2017 before Dye-2 May 2017 if table is supposed to be chronological. Include coordinates of sites and elevations in table. Possibly include radar units in table as well?*
We intended to have the table sorted alphabetical and chronological. You are right, since chronological comes first, we have to switch KAN-U up. Same appears for Table 2. We tried to include radar units but after including coordinates as suggested there is not enough space left for radar details other than antenna frequency.

*Figure 1 – Is OK how it is currently, but a little unclear - could take off elevation markings and put a color bar labeling elevations to see if it made the graphic a little less cluttered. Difficult to see the 500m marking near the coast. Not sure if the 1000,1500m elevations are labelled at all?*
We removed the contour labeling and included a colorbar for the elevation bands – as suggested.

*Table 2 – See comments from above, take out NASA-SE and EKT if they are not relevant for the rest of the study. Possible include depth of the pit? For the density ranges, could you use something other than "-" in the range column? It could confuse the reader if they did not scan the text for what the range meant.*
We changed the dash for the density ranges to "to". The column headline has been modified as well to clarify that density ranges are given. We kept EKT and NASA SE since they extent the presented density variation to a factor of 2 in accumulation.

*Figure 3 – Recommend using colors other than red and green stacked on one another for colorblind purposes. Could use a dashed line or a thicker line of another color.*
The respective color for KAN-U 2012/13 has been changed from green to purple to account for colorblind purposes. In case you are referring to Fig. 2, here, the red line has been changed to yellow to facilitate reading for colorblind persons.

*Figure 4 – Similar comment to above, try a different color than red and green since the lines are close to one another. Could use transparency to see the standard normal distribution behind.*
Fig. 4 has no green and red lines. We assume, you refer to Fig. 3. However, we increased the line width of the standard normal distribution lines in Fig.3 as well.

*Figure 4/ Figure 5 – Can these be combined into a 3 panel plot since you're talking about the same area?*
You are certainly correct and we attempted to combine those plots into one single figure. However, since TC will be printed as two-column paper, a smaller 1 panel plot will use less space than a 3 panel plot with a blank part underneath panel b.

*Table 3 – Never described what the major and minor axis are in the text for variograms. Does the mean prediction error need to be in the table if they are all 0 (expect the first value of 0.01)?*

In Section 2.3, we included: **After trend removal, we found directional anisotropy of the covariance in all of the longer transects, which means that accumulation variation varies with direction. Hence, we modeled variograms with different ranges per direction. In Table 3, we present major and minor axis of the range ellipsoid used for the variogram modeling.**

*Figure 9 – Need colorbar for the contour on 9a, difficult to see two locations of upGPR.*
We included the colorbar for the elevation bands and tried to facilitate visibility of the upGPR locations.

*Figure 4,6,7,9 – A personal preference is to have coordinates in lat/lon instead of UTM. If the scales do not allow though, especially for an area like Swiss camp, that is fine because it is on a few km scale.*
As you mention, the respective areas are rather small and, hence, we prefer the UTM grid to remain consistent for all figures.

*Other comments: The word "very" is used quite a bit throughout the manuscript as a qualifier and those instances can be removed the majority of the time. Using it does not add to the meaning of the sentences.*
Thank you for this suggestion. We checked whether the usage of the word "very" was necessary in the context of each sentence and removed/ changed expressions wherever useful.

---

## Author Comment (AC2) · 29 Nov 2019

We thank both referees and the associated editor for very constructive and helpful comments. There were several points raised by both referees that addressed similar or equivalent points. We listed the common points of criticism first before individual comments of each referee are considered separately. Minor changes such as typos have been incorporated in the MS without listing them here. In order to improve readability, comments by the respective referee are listed in italic, while responses and modifications in the MS are written in regular typesetting. Sentences and paragraphs being incorporated in the manuscript are listed in bold letters here and in the manuscript. To keep the manuscript up to date, we checked for recent publications and included some wherever appropriate. Within the introduction, we included Mottram et al. (2019) as another source for changes in mass loss processes and added Lewis et al. (2019) as another example of extensive ground-based radar campaigns. In addition, we exchanged the previously referenced Lewis et al. (2019) discussion paper in TCD to the now published Lewis et al. (2019) TC paper.

**Common points of criticism:**

- Both referees suggest to change the title of the manuscript. We decided to use the suggestion by Lynn Montgomery and changed the title to: **Relating regional and point measurements of accumulation in southwest Greenland.**

- Another point both referees criticize is the inconsistent/ interchangeable usage of SWE and snow accumulation within the manuscript. Surface mass balance (SMB) is solely used (and properly introduced, L27) within the introduction. Here, SMB is defined as …sum of snow accumulation and lateral redistribution by sublimation, wind and runoff…. This specifies the usage of the term "accumulation" and the importance of determining its spatial representativeness. In the revised manuscript, we consistently have changed the terminology to snow accumulation with symbol $b_s$ and units [kg/m$^2$].

- In addition, it has been suggested to simplify especially the section 2.3 dealing with spatial extrapolation. We now introduce terms such as variogram, nugget and anisotropy to facilitate readability of Section 2.3. Some radar terms are additionally explained as well.

- We modified the respective paragraphs in the introduction, which deal with objectives and scientific questions this work tries to answer. We fully agree that the main purpose of this manuscript is the relation of point measurements to regional accumulation. As stated by referee #1, the raised question (i) is a prerequisite to assess spatial representativeness and, hence, is removed from this listing. Since commonly applied in situ measurements of snow accumulations represent only a snapshot in time, it remains open whether accumulation patterns change with summer melt processes and are similar for two different winter accumulation season. We agree that the assessment of seasonal persistency cannot be properly determined with the available field data. However, since temporally continuous determinations of changes in accumulation are available and feasible in Greenland nowadays (upGPR, neutron probes), a relation of two consecutive years of data with point measurements is valuable and consequently is addressed in the results and discussion section. In addition, liquid water percolation has an effect on accumulation resulting in seasonal mass fluxes from the surface into deeper firn especially for

the investigated sites within the deep percolation zone of the Greenland Ice Sheet. We changed the respective paragraph to the following statement:

**The aim of this work is to relate point scales to regional scales of one to several square kilometers in area to improve our understanding of the representativeness of point measurements. For this purpose, we examine snow-pit and GPR data from two sites within the percolation zone of the GrIS and one site at the equilibrium line gathered over several field seasons. For each site, we investigate density variability between measurements from up to six snow pits within an area of 4 km² made in a single season, process radar transects of up to 25 km recorded in close proximity to those snow pits, and spatially extrapolate the radar-derived accumulation to estimate area-wide accumulation variability. For temporal comparisons, we use continuous observations of accumulation and melt recorded by upGPR \citep{Heilig2018}. Our results show that spatial representativeness of snow accumulation for a point measurement (snow pit) is high but values can be affected by local wind-induced surface roughness. We recommend to apply multiple snow depth measurements at the vicinity of the pits to better assess accumulation on regional scales.**

**Reply to referee #2:**

We highly appreciate comments raised by the referee and present a point-to-point reply for all issues raised by the referee. For an improved readability and to facilitate direct response, we sometimes subdivided comments into several paragraphs referring to similar issues Please also note our general response at the top of this document.

*This paper tries to answer the question of how representative point measurements of snow accumulation are for the larger regional scale. This subject is important, urgently needs attention, and this paper fills a void in our knowledge on the connection between the observation scale and the (regional) climate modelling scale. Scientifically, the paper is solid, and I have few methodological remarks. In terms of presentation however, I have quite a few remarks. Changing a word or sentence here or there won't fix the fact that the paper is quite tough to read.*

We thank the referee for the evaluation and the overall positive assessment.

*General*

*- Various terms are used interchangably, without a proper definition. Snow accumulation, SMB, SWE, snowfall, snow depth. Please have a critical look at the terminology, simplify, and make uniform.*

Please see the common comments above. We agree that snow accumulation and SWE were used inconsistently. Snowfall and snow depth are standing terms all described in Fierz et al. (2009). We do not consider it being necessary to introduce these terms. SMB is only used within the introduction where it is properly introduced.

*- I had to dig quite deep in my memory to connect the dots between variograms,nuggets, space-invariance, isotropy and stationarity. Would it be feasible to ease the text in section 2.3?.*

A criticism raised by referee #1 as well. We now introduce each geostatistical term within this section. Range in the sense of correlation range is consistently used as correlation range from now on.

*- The abstract is particularly awkward in grammar and style, as if it was the last part that was written and not checked before submission. I'll give three example sentences and how to make this readable:.*

We sincerely apologize for the sloppiness of the abstract and carefully revised the entire abstract. We included all recommendations and now hope it is significantly simplified.

**In recent decades, the Greenland ice sheet (GrIS) has frequently experienced record melt events, which significantly affected surface mass balance (SMB) and estimates thereof. SMB data are derived from remote sensing, regional climate models (RCMs), firn cores and automatic weather stations (AWSs). While remote sensing and RCMs cover regional scales with extents ranging from 1--10~km, AWS data and firn cores are point observations. To link regional scales with point measurements, we investigate the spatial variability of snow accumulation ($b_s$) within areas of approximately 1—4 km$^2$ and its temporal changes within two years of measurements. At three different sites of the southwestern GrIS (Swiss Camp, KAN-U, Dye-2), we performed extensive ground-penetrating radar (GPR) transects and recorded multiple snow pits. If the density is known and the snowpack dry, radar-measured two-way travel time can be converted to snow depth and $b_s$. We spatially filtered GPR transect data to remove small scale noise related to surface characteristics. The combined uncertainty of $b_s$ from density variations and spatial filtering of radar transects is at 7--8\% per regional scale of 1—4 km$^2$. Snow accumulation from a randomly selected snow pit is very likely representative of the regional scale (with probability p=0.8 for a value within 10\% of the regional mean for KAN-U, and p>0.95 for Swiss Camp and Dye-2). However, to achieve such high representativeness of snow pits, it is required to determine the average snow depth within the vicinity of the pits. At Dye-2, the spatial pattern of snow accumulation was very similar for two consecutive years. Using target reflectors placed at respective end-of-summer-melt horizons, we additionally investigated the occurrences of lateral redistribution within one melt season. We found no evidence of lateral flow of meltwater in the current climate at Dye-2. Such studies of spatial representativeness and temporal changes in accumulation are necessary to assess uncertainties of the linkages of point measurements and regional scale data, which are used for validation and calibration of remote sensing data and RCM outputs.**

*- In several parts, you claim that snow accumulation should be established for an area of at least 20 x 20 m. This seems a very important implication for future field work. However, I miss the quantitative underpinning of these numbers. Why not 10 x 10, or25 x 25 m? And how should this be done if no GPR is available? This is such a crucial part of the manuscript that I expect some more discussion of the implications on field practice.*

We included an analysis on benefits from multiple snow probings on the assessment of the mean snow depth per area. This changed a large fraction of the respective section:

**The above results imply that a point measurement of $b_s$ (snow pit, upGPR value, neutron probe, etc.) is representative for an area of roughly 4x4 km$^2$ at Dye-2 with a probability of p ≥ 0.9 and an uncertainty of ±10% in case snow depth is averaged. For KAN-U, the spatial variability is slightly higher and, consequently, there is less certainty about how well a single measurement represents the surrounding area. However, we consider a probability of p ≥ 0:8 with uncertainty of ±10% for both study sites as a resilient estimate.**

**To quantitatively assess the benefit of snow depth measurements in addition to a snow pit, we numerically assume a sinusoidal snow depth variation with wavelengths of 56 m (arithmetic mean of the previously presented range in wavelength for the GPR transects) and average amplitude of ±6.8 cm (the fluctuations in snow depth from arithmetic mean). Averaging multiple snow depths (with a sampling distance of 1 m) from a 20 m long probing transect, result in a maximum possibly measured offset in snow depth of -20\% (amplitude decreases to 5.4 cm). A 10 m long probing line reduces the maximum offset by -6\% compared to single point measurements (6.4 cm amplitude). A 30 m long snow probing line, however, result in a decrease of maximum possible offsets by -44\% (3.8 cm**

**amplitude). An additional cross line of probings will further decrease offsets. Only if the surface features are aligned symmetrically in both probing directions, the maximum offset derived from both lines will theoretically remain stable. For a measured snow pit with $\rho_s$=350 kg/m$^2$ and $L_s$=1 m, the combined regional uncertainty (±5\% density uncertainty, ±6.8 cm snow depth variation) reduces from a single point measurement with $b_s$ = 350±42 kg/m$^2$ to a maximum possible uncertainty of $b_s$ = 350±35 kg/m$^2$ for just a single 20 m probing line. These numerical results confirm values for representativeness derived from geostatistical extrapolation. Hence, we recommend to combine a larger number of snow-depth probings within an area of at least 20 m by 20 m in the vicinity of the pits to increase the regional representativeness. Regional snow density variations of ±5\% can be accepted if snow depth uncertainty is minimized. Snow probing lines can easily be performed with respectively low time consumption compared to multiple snow pits. In particular, the wind-induced surface roughness has to be accounted for to provide spatially-representative $b_s$ values.**

*- The title is inappropriate. My suggestion would be: "Representation of point measurements for regional-scale snow accumulation in/on the southwestern Greenland Ice Sheet."*
See above, the title has been modified in accordance to Lynn Montgomery's suggestion and we believe, it addresses your concerns as well.

*- Throughout the paper, you seem to use rho_s mostly as a bulk parameter: a mean over a certain depth. Can you more clearly distinguish between the actual snow density and this vertically integrated bulk density, and define the bulk density clearly?*
We now specify "bulk snow density" when it first appeared in the methodology section: **In dry snow and firn (with two contributing volume fractions $\theta_a$+$\theta_i$=1), the wave propagation depends solely on the relation of air ($\theta_a$) to ice volume fraction ($\theta_i$) (e.g., \citealp{Kovacs1995,Maetzler1996}). Hence, with the bulk snow density ($\rho_s$, the average density of the entire snow column) measured in snow pits, we can convert from TWT to snow depth ($L_s$) and the amounts of bulk accumulation $b_s$ with unit kg/m$^2$) using the equation**

*Specific remarks*
*- Title: "South-Western". I looked it up but this should be either "southwestern Green-land" or "Southwest Greenland".*
Title has been changed - see above.

*- L1: significant changes. In what?*
See above the abstract has been changed significantly.

*- L3: remove the sentence "Data sources ... coverage"*
Sentence has been removed.

*- L3: remove "at least"*
Changed accordingly.

*- L11: "per regional scale"?*
Changed to: **The combined uncertainty of $b_s$ from density variations and spatial filtering of radar transects is at 7--8\% per regional scale of 1—4 km$^2$.**

*- L11: "to analyze for". To analyze is a transitive verb. Suggestion "we investigate". This recurs frequently in the text (e.g. P1L17)*

We thank the referee for highlighting this. We haven't been aware that analyze is a transitive verb. We consistently substituted "to analyze" with "to investigate" or verbs with similar meaning, where appropriate.

*- L18: "cannot be evidenced" -> "At Dye-2, we found no evidence of..."*
Changed accordingly.

*- L27: SMB related processes -> SMB*
*- L35: "certainly contribute" -> "may also contribute"*
*- L52: remove "at the GrIS"*
*- L57: point -> location*
All changed accordingly.

*- L70: suggest "To improve our understanding of the representativeness of ..."*
Has been changed and rephrased to:
**Point observations, such as snow pits and ice cores are usually performed once a year at most. Such temporal snapshots limit the evaluation of spatial representativeness as they can be influenced by recent weather conditions. Hence, it is necessary to clarify whether regional accumulation patterns are consistent over more than one accumulation season to investigate if temporally continuous point measurements such as AWS data, upGPR and neutron probes remain representative.**

*- L71: "in area for two sites" -> " in areas around two sites"*
We have removed the respective sentence.

*- L72: I have once been taught that one paper answers one major question. Your one major question is about representativeness of point measurements. All other questions are hurdles that you come across while answering that question. My suggestions would be to rephrase, and to formulate L70-80 such that you introduce the different steps needed to answer your "major question" with associated sections.*
We rephrased the respective paragraph as listed above.

*- L72: "equilibrium line of altitude" -> "equilibrium line"*
*- L125: numerous -> several*
*- L144: range -> ranges*
*- L146: constantly -> always*
*- L172: more constraint -> better constrained*
*- L181: remove "a realm of"*
All changed accordingly.

*- L197: Do not start a sentence with a mathematical symbol.*
Modified to: The term $b_{s,N}$ is simply…

*- L196 - 206: past and present tense are used here interchangably. Please unify.*
We apologize for this. It is now unified.

*- L217: snow depths -> snow depth*
Changed accordingly.

*- L217: why call this ice volume fraction? Suggestion to simplify these sentences:*

*"We investigate the error that we introduce by assuming a single bulk density in the conversion from TWT to snow depth for an entire GPR transect. For that, we use a collection of snow pits, several from each of five locations, that were collected in a period of three years (table 2)."*

Modified to: **We investigate the error that we introduce by assuming a single bulk density in the conversion from TWT to snow depth for an entire GPR transect. Hence, we determine the spatial variability in density within the respective area. Table 2 presents snow-pit data from our three study sites and two additional sites.**

*- L243: above average -> above-average (idem below-average)*
*- L327: larger scale -> larger-scale*
*- L341: north to south direction -> north-to-south direction*

We have not yet inserted dashes for all such phrases. Such details are very specific and treated differently depending on the style and language of each journal. If applicable such phrases will be corrected within the final editing phase with the journal directly.

*- L261: equals to -> equals*
Changed accordingly.

*- Figure 4: consider inverting the color scale. Blue = low accumulation, yellow is high accumulation*
Here, we respectfully disagree. We consider it being more intuitive to have high accumulation associated with blue color and low accumulation with yellow.

*- L293: awkward construction. Rephrase*
Modified to: **The unfiltered data, however, show a decreased representativeness with p=0.89 in 2015/16 and p=0.77 in 2016/17 for the same uncertainty range of ±10\%. Here snow depth is solely derived from the snow pit. Such values demonstrate that $b_s$ data derived simply from a snow pit without averaging snow depth around the pit location will decrease the area-wide representativeness at Dye-2.**

*- L316: Not all of the collected radar transect patterns (grids?) ...*
Sentence has been changed to: **Not all of the recorded radar transect grids are ideal for the applied geostatistical analyses.**

*- L347: this sentence is not complete*
The whole paragraph has been modified as suggested by referee#1. KAN-U is no longer used for this analysis.
**To quantitatively assess agreement in accumulation patterns, we used the respective normalized accumulation data and calculated the quotient. The cumulative data distribution of the quotients is presented in Figure 8. A constant area-wide quotient of 1 would imply that the scaled accumulation patterns are exactly equal. For Dye-2, the probability of data being equally distributed in May 2016 and 2017 with a given uncertainty of ±10\% is p≥0.95, meaning all intersecting locations of the accumulation pattern in two consecutive years at Dye-2 are similar.**

*- L368: LaTeX error*
Corrected.
*- L387: 40% -> 40% lower*
Corrected.

*- L408: The conclusion about persistence is unsatisfactory, and you seem to be shifting goal posts in the manuscript. In the abstract you write that interannual accumulation patterns "are very persistent". In section 3.3, the 2016 and 2017 data are "very similar". Then in L408 you say that "results suggest persistence". I think you should refrain at all from inferring persistence based on two data points. It's ok to mention that the patterns were similar in both 2016 and 2017, but I don't think there is enough argument here to start discussing persistence.*

We fully agree and weakened consistently throughout the manuscript pattern persistence to changes in accumulation pattern for 2016 and 2017 or agreement in accumulation patterns. In the conclusion, we state:

**Our results suggest that there is only little change of accumulation patterns at Dye-2 for spring 2016 and 2017.**